# Integrating functional connectivity in designing networks of protected areas under climate change: A caribou case-study

Sarah Bauduin[1,2], Steven G. Cumming[1], Martin-Hugues St-Laurent[3]*, Eliot J. B. McIntire[1,4]

1 Faculty of Forestry, Geography and Geomatics, Laval University, Quebec, Quebec, Canada, 2 CEFE, CNRS, EPHE, University of Montpellier, IRD, Montpellier, France, 3 Département de Biologie, Chimie et Géographie, Center for Northern Studies, Center for Forest Research, Université du Québec à Rimouski, Rimouski, Quebec, Canada, 4 Canadian Forest Service, Natural Resources Canada, Victoria, British Columbia, Canada

* martin-hugues_st-laurent@uqar.ca

**Data Availability Statement:** Data cannot be completely shared publicly because of the Endangered Status of the Atlantic-Gaspésie caribou population, so data that could help track and find

## Abstract

Land-use change and climate change are recognized as two main drivers of the current biodiversity decline. Protected areas help safeguard the landscape from additional anthropogenic disturbances and, when properly designed, can help species cope with climate change impacts. When designed to protect the regional biodiversity rather than to conserve focal species or landscape elements, protected areas need to cover a representative sample of the regional biodiversity and be functionally connected, facilitating individual movements among protected areas in a network to maximize their effectiveness. We developed a methodology to define effective protected areas to implement in a regional network using ecological representativeness and functional connectivity as criteria. We illustrated this methodology in the Gaspésie region of Québec, Canada. We simulated movements for the endangered Atlantic-Gaspésie caribou population (*Rangifer tarandus caribou*), using an individual-based model, to determine functional connectivity based on this large mammal. We created multiple protected areas network scenarios and evaluated their ecological representativeness and functional connectivity for the current and future conditions. We selected a subset of the most effective network scenarios and extracted the protected areas included in them. There was a tradeoff between ecological representativeness and functional connectivity for the created networks. Only a few protected areas among those available were repeatedly chosen in the most effective networks. Protected areas maximizing both ecological representativeness and functional connectivity represented suitable areas to implement in an effective protected areas network. These areas ensured that a representative sample of the regional biodiversity was covered by the network, as well as maximizing the movement over time between and inside the protected areas for the focal population.

**Funding:** Funding for this project was provided by: - the Québec Fonds Verts (S.G.C., E.J.B.M.) (URL: http://www.environnement.gouv.qc.ca/ministere/fonds-vert/) - the Ouranos Consortium (S.G.C., E.J.B.M.) (URL: https://www.ouranos.ca/) - the Centre d'étude de la forêt (S.B.) (URL: http://www.cef-cfr.ca/) - Natural Sciences and Engineering Research Council Discovery Grants (E.J.B.M., S.G.C.) (URL: https://www.nserc-crsng.gc.ca/NSERC-CRSNG/) - the Canada Research Chair's program (E.J.B.M.) (URL: https://www.chairs-chaires.gc.ca) -the French National Research Agency (ANR-16-CE02-0007, S.B.) (URL: https://anr.fr/en/).

# Introduction

Habitat change is recognized as the main driver of the current declines of terrestrial species [1–3]. Securing habitats by creating or expanding protected areas networks is part of the solution to counter biodiversity loss [4]. A regional protected areas network could be considered ultimately effective insofar as it can sustain the region's biodiversity into some reasonably foreseeable future. Such effectiveness is not guaranteed [5, 6] and could be limited by many factors [7]. We consider three such factors: ecological representativeness [8], functional connectivity among protected areas within the network [9] and resilience to climate change [10].

Ecological representativeness measures the degree to which the various non-anthropogenic habitats or ecosystem types (*sensu* [11]) within a focal region are available within a protected areas network in proportion to their regional abundance [12, 13]. When protected area locations are skewed towards certain habitats [14, 15], usually for economic or social reasons, ecological representativeness will be low, and habitats considered to be of high-value for some species may be underrepresented [8]. Conversely, when ecological representativeness is high, it is reasonable to assume that the habitat requirements for many species will be satisfied within the protected areas network. This assumption is usual in conservation, and representativeness is one of the core concepts in systematic conservation planning [13, 16].

A high degree of ecological representativeness may be a necessary condition for an effective protected areas network, but it is not sufficient. There are species whose requirements are not automatically satisfied by ecological representative networks, such as endemic or threatened species [17]. Another exception, which we explore here, would be wide-ranging species with habitat requirements varying among seasons or life history stages. Functional connectivity is "*the degree to which the landscape facilitates or impedes movement among resource patches*" [18] or, in this case, among protected areas within a protected areas network. Functional connectivity is species- or population-specific [19, 20]. A protected areas network with high functional connectivity facilitates the movement between different protected areas for individuals of a given species, increasing their access to resources and, ultimately, the rates of recruitment or survivorship [21]. Increasing functional connectivity may thus increase population size and decrease extinction risk for the vulnerable species [21]. These effects would increase the effectiveness of a protected areas network.

Climate change is a major driver of ecosystem change and its negative impacts on biodiversity have increased rapidly over the past century [1, 3, 22, 23]. Climate change disrupts environmental patterns and species' habitats globally [24, 25]. As a result, species distributions and individual movement patterns are impacted [26, 27]. Because of the "cost of waiting," managers should proactively account for future climate change effects when implementing new protected areas networks [28] or expanding existing networks. The effectiveness of a fixed protected areas network designed for current conditions may decrease in the future if ecological representativeness or functional connectivity decline. Enhancing or maintaining functional connectivity inside protected areas networks is thus one approach to helping species cope with climate change [10, 29]; individuals would be better able to access resources available in distant protected areas, or even to migrate from less to more favorable areas. Therefore, the ability of a protected areas network to sustain functional connectivity under climate change is another dimension of effectiveness.

Many methods exist for designing protected areas networks to achieve ecological representativeness. In the systematic conservation planning literature [13], variations of the site selection problem are posed, where one seeks a subset of available sites that, in aggregate, achieve some measure of ecological representativeness at a near-minimum of total area or cost. Variations of these approaches exist that can also partially satisfy other conservation objectives such

as topological connectivity, where selected sites are spatially aggregated to some degree [e.g. 30].

The aim of our paper is to present a new method to include functional connectivity in the design process, while also accounting for its persistence under climate change. Our method applies individual-based movement models to simulate maps of habitat use by a population, under present conditions and under hypothetical future conditions reflecting climate change scenarios. From these maps, we derive numerical indices of functional connectivity that allow alternate protected areas network designs to be compared. These are coupled with a novel variant of the site-selection algorithm that constructs protected areas networks of specified total area that can achieve a high degree of ecological representativeness while also satisfying secondary constraints on hydrological connectivity and intactness; the variant is detailed in Schmiegelow et al. [31] and an example of application can be consulted in Saucier [32].

Our goal was to identify protected areas network designs that simultaneously achieve high degrees of ecological representation and functional connectivity under present conditions and future climates. Achieving this is a multi-objective optimization problem [33, 34] that does not admit a unique solution. Instead, one may define a tradeoff surface by those points or "feasible solutions" where no single objective can be increased without decreasing at least one other. Such points are said to be Pareto optimal; points on the interior of this surface are suboptimal in that other solutions exist that are better in terms of all the dimensions considered [35]. Our design methodology used randomization to generate a large sample of feasible solutions for a given protected areas network design problem. Using these samples, we could approximately define the tradeoff-surface and identified a set of feasible, near-equivalent solutions in the vicinity of any specified point on that surface.

We illustrated our method in the Gaspésie region (Québec, Canada) using the endangered Atlantic-Gaspésie population of woodland caribou (*Rangifer tarandus caribou*) as our focal population. This isolated herd, which has been identified as one of 12 Designatable Caribou Units considered irreplaceable components of Canada's biodiversity [36], has been declining since the late 19th century and has reached a critical abundance level [37]. To measure functional connectivity with respect to this population, we applied an individual-based model of animal movement previously developed for this population [38]. The Québec government is currently engaged in expanding the existing network of protected areas in this region to attain a proportional area target of 12% [39]. To place our work in the context of this ongoing conservation planning exercise, we employed a variant of our design methodology that constructs protected areas networks by adding new protected areas to the existing network. We identified areas that were of high relative importance in achieving ecological representation in the Gaspésie region and functional connectivity for the Atlantic-Gaspésie caribou under present conditions and future climates.

## Methods

### Overview

We present a method to identify potential areas to prioritize to create an effective protected areas network in a given region, conserving both the regional biodiversity, represented by biophysical surrogates as well as focal, highly mobile endangered species, while also accounting for climate change. We illustrated this method in the Gaspésie region of Québec, Canada (Fig 1) using an individual-based model we developed previously [38] and for which we used telemetry data collected on several individuals. We had an Animal Welfare certificate (#52-13-112) for the capture and manipulation of these caribou, but for the current manuscript, no additional capture session was conducted.

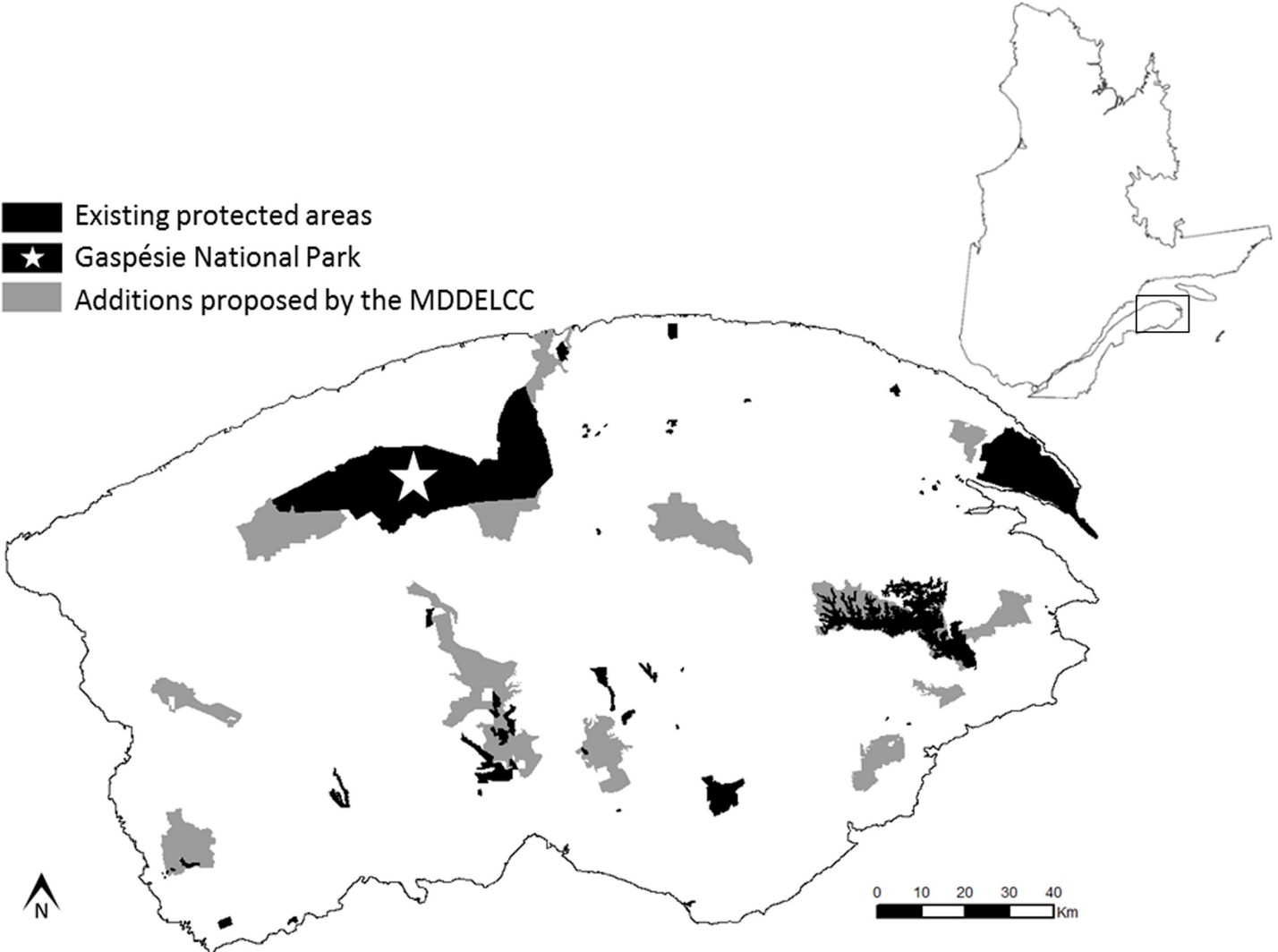

**Fig 1. The Gaspésie natural region with existing protected areas (black) and additions (grey) proposed by the Québec government [39].** The Gaspésie National Park (i.e. the black area designated by a white star) encompass most of the range and all the breeding habitats used by the Atlantic-Gaspésie caribou population. Right inset: Province of Québec (Canada) with study area outlined.

We built candidate protected areas (hereafter referred to as CPAs) in the region and integrated random subsets of these with the existing protected areas (Fig 1) to create a large sample of candidate networks of the desired total area. Each candidate network was then evaluated for effectiveness. High priority conservation areas were identified as the CPAs that occur most frequently in the most effective networks. The complete workflow is summarized in Fig 2.

## Study area

The Gaspésie natural region (latitude extent: 47.98 to 49.20º N, longitude extent: 64.11 to 67.57º W; Fig 1) is a physiographically defined area of approximately 25,000 km$^2$ at the eastern end of the Gaspésie Peninsula, in eastern Québec [39]. Except for narrow coastal bands, it belongs to the balsam fir–white birch bioclimatic domain [40]. The climate is maritime with abundant precipitation. Wildfire is infrequent; the main natural disturbance is spruce budworm (*Choristoneura fumiferana*) outbreaks [40]. Approximately 90% of the region is covered

## Delineate Hydrological Catchments

Define catchments from the drainage network and a digital elevation model.

Calculate catchment intactness

Calculate ecological indicators:
1. Mean elevation
2. Class frequencies of categorical variables

Select intact headwater catchments as seeds.

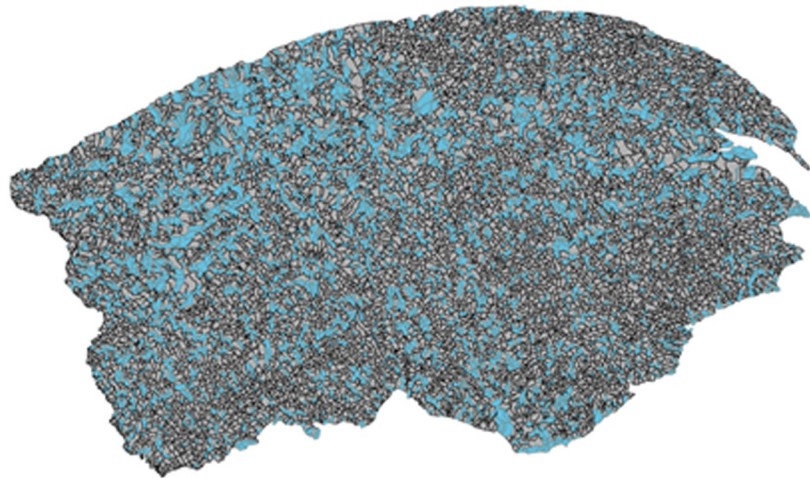

## Construct Candidate Protected Areas

From each seed catchment, traverse the stream network adding intact catchments until the size criterion is met. The results are the Candidate Protected Areas (CPA).

The total area covered by the 690 unique CPAs is shown on the right (overlaps not shown).

Mean CPA size was 86.8 km².

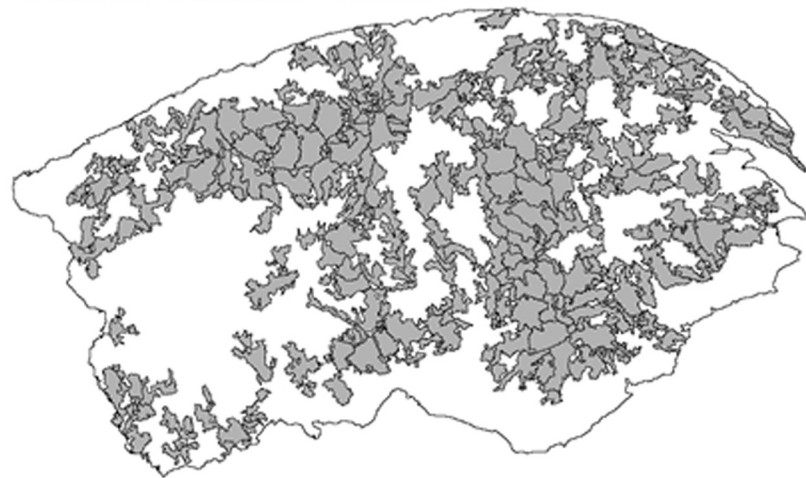

## Sample and Evaluate Candidate Networks

Generate a set of Candidate Protected Areas Networks (CNs) by adding random sequences of CPAs to existing protected areas, up to the total area target of 3,080 km².

Evaluate ecological representation and functional connectivity indices for each CN.

Generate a sample of CNs approximating some tradeoff between the indices, as illustrated in Figures 3 and 4; one resultant CN is shown at right.

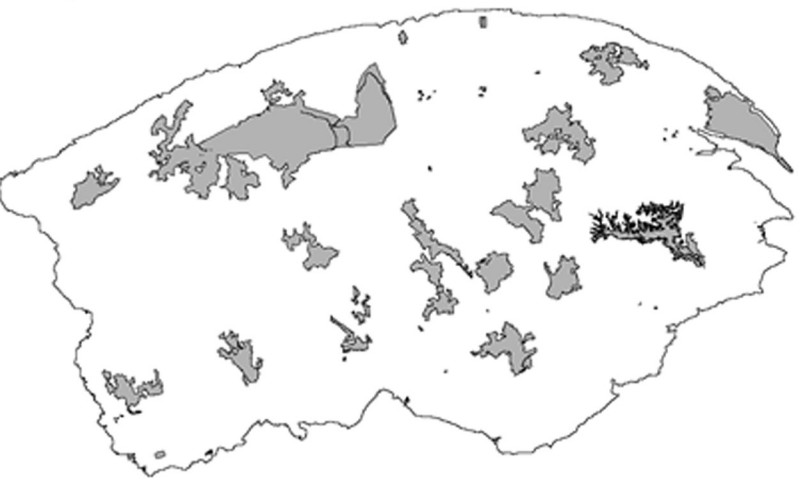

**Fig 2.** Schematic of workflow for assembling candidate protected areas networks (bottom panel) from candidate protected areas (middle panel), which are constructed as hydrologically connected groups of intact hydrological catchments (top panel) rooted at the headwaters (shown in blue).

by forests and 80% of these are on public lands [39]. Forest harvesting and the associated extensive road network are the main proximate human disturbances to date. Only 34% of the Gaspésie natural region is free of measurable human footprint [39].

An existing protected areas network covers 5.5% (1371 km$^2$) of the region (Fig 1). This falls short of the target of an ecologically representative 12%, set by the government of Québec [41]. A scenario proposed by the Ministère du Développement Durable, de l'Environnement et de la Lutte aux Changements Climatiques ("the Ministry", henceforth) defined 20 new protected areas that would increase the percentage of area protected to 12.3% (3080 km$^2$) [39; Fig 1].

The Gaspésie National Park (802 km$^2$) is the largest protected area in the region [42]. This park helps conserve 42 endangered and vulnerable species of plants and animals [39], including the Atlantic-Gaspésie caribou population. Because of its designated status as endangered [43], its capacity to move widely [44] and the vulnerability of its habitat [45], we chose this caribou population as our focal population to define functional connectivity. The population is designated because of its small size (currently ~70 individuals [37]) and the observed long-term decline [46]. These caribou rely on alpine tundra [47, 48] as a predator-free refuge against coyotes (*Canis latrans*) and black bears (*Ursus americanus*) [49], a habitat possibly threatened by climate change [50, 51]. Caribou also use mid-elevation, old fir forests during winter [47,48]. The fir forests around the park are affected by forest harvesting, and also possibly by climate change [52].

## Building protected areas networks

We adapted the BEACONs (Boreal Ecosystems Analysis for Conservation Networks) approach to create protected areas networks by adding new protected areas to the existing ones [31, 32]. This approach uses mapped hydrological catchments as spatial units, rather than using the cells of an arbitrary grid or other tessellation (Fig 2). This allows the design of protected areas that simultaneously achieve terrestrial and hydrological connectivity [53, 54]. Although functional connectivity for aquatic species was not a major concern in our study area [39], it is increasingly recognized as a desirable goal in general [55]. The use of hydrological catchments does impose some limits on the shape and size of protected areas, due to the size of catchments and the method of stream network traversal; however, the size of the catchments was small, implying high spatial resolution to the design. CPAs are assembled as contiguous, hydrologically connected sets of catchments satisfying minimum size and intactness criteria (Fig 2). Using an initial "seed" catchment as starting point, a modified breadth-first traversal of the stream network is carried out. Catchments are added as they are encountered, provided they satisfy a catchment-level intactness criteria. The process continues until the target size is achieved, or all pathways are blocked by headwater or non-intact catchments.

We used a custom 1:50,000 catchment layer for Gaspésie [56] (Fig 2). The catchment's average size was 2.3 km$^2$ (SD = 1.3 km$^2$). We defined catchment intactness using existing 250 x 250m raster maps of the human footprint in Gaspésie [39]. These maps defined six disturbance types: forestry activities, roads and trails, agriculture, power-line rights-of-way, urban areas, and "other". Only the first two of these disturbance types were widespread in our study area. From the six disturbance rasters, we derived a 250 x 250m raster of cell-level intactness, using a value of 0 for cells where no disturbance of any type was present, and a value of 1 otherwise. Catchment-level intactness was then calculated as the mean cell-level intactness over all cells within a catchment, or equivalently, as the proportion of non-intact cells. We defined our catchment-level intactness criteria as the median catchment intactness over all catchments on public lands, outside of existing protected areas (median value = 0.026) (Fig 2). In order to create feasible designs given land ownership in Gaspésie, we set the intactness of catchments

completely overlapping private lands [39] to zero so that they would be excluded from the CPA construction (Fig 2).

Our measures of cell and catchment-level intactness give equal weight to each disturbance type. This was not applied, in general, at the species level. The response to different disturbance types is known to differ among species, as shown by Toews et al. ([57]) in Alberta for wolves (*C. lupus*), meso-carnivores and large ungulates, or more precisely within our study area for caribou, coyotes and bears that respond differently to recent harvesting, forest roads and hiking trails [58]. However catchment intactness as a design criteria was intended to be relevant to biodiversity more broadly, not to any particular species. Equal weighting of disturbance types reflects different responses to disturbances across species, and is thus a conservative assumption.

We prioritized the inclusion of headwater catchments in protected areas, because of their hydrological importance and to reduce the potential for upstream river contamination inside protected areas [59]. Accordingly, we used all intact headwater catchments as seeds. We used the mean size of the new protected areas proposed by the Ministry (Fig 1; mean = 85.5 km$^2$, SD = 61. 4 km$^2$; [39]) as the target size. We then applied the construction process to each seed catchment, with target size and catchment intactness criteria as defined above. The procedure returns a list of the catchments selected and their total area. If this total area satisfied the target size, the result was accepted as a candidate protected area. The construction adds entire catchments one at a time, so target sizes are generally exceeded rather than satisfied exactly (Fig 2). Because the mean catchment area (2.3 km$^2$) was small relative to the target size, we consider such discrepancies negligible. The minimum size of the Ministry's proposed new protected areas was 31.2 km$^2$ [39], which is only 36.5% of our target size. To make our size criteria more comparable to theirs, we also accepted, as candidate protected areas, constructions smaller than the target size but larger than this minimum. Candidate protected areas satisfy, by construction, the criteria of size, intactness and hydrological connectivity [31]. They are not designed to satisfy ecological representativeness or functional connectivity criteria. These criteria pertain to *networks* of protected areas, not to individual protected areas.

We used the set of selected candidate protected areas to generate a sample of 500,000 candidate protected areas networks (Fig 2). Each candidate network was constructed by adding a random sequence of candidate protected areas to the existing network, until the summed area of unique catchments exceeded the Ministry's area target of 3080 km$^2$ (Fig 2). The candidate protected areas included within a given candidate network may overlap with each other or with elements of the existing network, but this does not affect the candidate network area. It is easy to show that the properties of hydrological connectivity and intactness are conserved by aggregating overlapping candidate protected areas. Thus, the only material consequence of such overlap is that spatially disjunct network elements may exceed the minimum size criteria, and may be fewer than expected based on the number of candidate protected areas that were added. However, other things being equal, larger protected areas are preferred [60], and the number of spatially separated components is not one of our network design criteria, so we do not consider this overlap of relevance to the present study.

## Network ecological representativeness

Ecological representativeness is measured at the candidate network level [31], in terms of four environmental attributes: elevation, surficial deposit, drainage class and potential vegetation type [39] (Fig 2). Potential vegetation was defined by the Ministère des Forêts, de la Faune et des Parcs du Québec (MFFP) as the vegetation present on a site or potentially present, in the absence of disturbance [41]. Surficial deposit, drainage class and potential vegetation type had

been used for a gap analysis conducted to inform the proposed expansion of the existing protected areas network [39]. The four attributes define habitats independent of human activities such as harvesting history. All four attributes were available for the entire study and were provided by the Ministry. Catchment-level attributes were calculated by intersecting a shapefile of the catchment layers with the various raster grids and taking means for continuous attributes and frequency tables for categorical attributes.

We measured candidate network representativeness using nonparametric, two-sample univariate dissimilarity measures. For each attribute, we obtained its distributions over all catchments within the candidate network, and over all other catchments within the study region. For continuous attributes (e.g., elevation), we calculated a two-sample Kolmogorov-Smirnov statistic. For categorical attributes (e.g. surficial deposit, drainage class, and potential vegetation type), we calculated the Bray-Curtis statistic. These statistics ranged from 0 to 1, measuring a candidate network's deviation from perfect representation with respect to an attribute, and were equal to 0 only when the two distributions are identical. The univariate statistics for multiple attributes were combined into a univariate distance metric by calculating a Euclidean norm [31, 32]. We took the inverse of this distance metric as the ecological representativeness score for each candidate network, so that a larger score indicates a greater degree of representativeness.

We also calculated the representativeness score of the Ministry's proposed network. To do this, the protected areas within this network were approximated to the resolution of hydrological catchments. Because these catchments were relatively small, we assumed approximation errors to be negligible. We note that this measure of ecological representativeness does not depend on the specific choice of ecological attributes or on the specifics of our design methodology; it can be applied to any existing protected areas network using whatever ecological attributes are of interest.

## Network functional connectivity

We defined candidate network functional connectivity for the Atlantic-Gaspésie caribou population using a spatially explicit individual-based movement model (IBM) adapted from Bauduin et al. [38, 61] (Fig 2). This model simulates caribou movement as a two-state behavioral model process with a habitat-mediated random walk in habitats of high quality and a foray loop movement in habitats of low quality. The IBM includes the attraction of individuals' mating areas during mating season. Habitat quality is modeled using a Resource Selection Function (RSF; [62, 63]) model developed for this specific population of caribou [58] and based on four habitat types relevant to this population (i.e., alpine tundra, mature fir stands, regenerating stands and stands of other tree species, primarily broad-leaved species) as well as three classes of transportation routes (paved roads, gravel/secondary roads, and hiking trails). These landscape components represent resources or barriers for the caribou (e.g., alpine tundra and mature fir stands for food resources and shelter from predators, paved roads as movement barriers) or proxies for the presence of their predators (e.g., regenerating stands where bears and coyotes frequently occur).

The original model was re-estimated from newly available GPS locations [44, 64]. Model movement behavior is partially driven by spatial variation in habitats as described by the RSF. Following Bauduin et al. [38, 61], we generated five different habitat maps to represent the current landscape and four landscapes for 2080 forecast under different climate change scenarios. The current landscape was derived from the ecoforestry maps from Québec's 4th decennial forest inventory program (source: MFFP). The future landscapes represented a range of potential climate change impacts (none: $CC_0$; minimum: $CC_{Min}$; moderate: $CC_{Med}$; high: $CC_{High}$) on

vegetation dynamics and natural disturbances, combined with predicted cumulative timber management for 2080. Primary impacts were reduced areas of tundra [50, 65] and of fir forest [52], and a decreased severity of spruce budworm outbreaks [66, 67] leading to less area with young fir stands. A complete description of the four climate scenarios and of the construction of the corresponding 2080 landscapes is given in Bauduin et al. [61] (and in the S1 Material). A fifth scenario represented current conditions.

For each landscape, we created 20 individuals in each of the three caribou subpopulations (see [38]; current population estimates are 20, 35 and 15, [37]) and ran 10,000 replicates of four-year model simulations. For each scenario and replicate, we calculated and mapped the number of caribou visits per 1-ha landscape cell (see S2 Material). For each candidate network, under each scenario, we measured network functional connectivity by taking the mean, over replicates, of the number of caribou visits in cells within the network's protected areas. This represented the simulated movement patterns within and between the protected areas. The differences in results between climate scenarios were relatively small. We considered evaluating differences between them to be unimportant relative to our core message concerning the inclusion of functional connectivity in a protected areas network design. Accordingly, and because no climate change scenario was defined as more likely than another, we used the mean functional connectivity over the four climate change scenarios as an index of network connectivity under future conditions (S2 Material). We calculated the functional connectivity of the Ministry's network under current and future conditions in the same way.

## Identifying priority conservation areas

By plotting the distributions of indicators (i.e., ecological representativeness, current and future functional connectivity), the outlines of the tradeoff surface may be visualized. To identify one specific location on this surface, we incrementally decreased a quantile threshold Q from 1 to 0 by steps of 0.001 to find the maximum Q for which at least 500 networks had indicator values above the Q-th quantile for each of the three indicators. This identified a subsample of 0.1% of the candidate networks that were highly ranked under all three criteria. This represents a level of tradeoff which, informally, values all three attributes equally. We then identified the candidate protected areas included in the network subsample, calculated their selection frequencies, and mapped their locations color-coded by selection frequency. These selection frequencies may be interpreted as priorities for including candidate protected areas within a new or expanded protected areas network [30]. Similarly, spatial clusters of high priority areas may highlight regions of high importance relative to network conservation goals. We set a selection frequency threshold for high priority catchments by plotting the selection frequencies in decreasing rank order, and identifying an inflection point on this curve. We then added the locations of high priority catchments to the above-mentioned map. To evaluate the sensitivity of the priority conservation areas selection to the choice of conservation goals, we applied the preceding analysis to two alternate rankings of the same set of 500,000 candidate networks. In the first instance, we ranked the networks by ecological representation alone. Then, we ranked them according to their joint functional connectivity under current and future conditions.

## Results

### Protected areas networks

We constructed 690 unique candidate protected areas satisfying our size and intactness criteria. Their mean size was 86.8 km$^2$ (SD = 4.0 km$^2$), only very slightly larger than the target size of 85.5 km$^2$. The 500,000 candidate protected areas networks (candidate networks) included a

mean of 25 (SD = 2.2) candidate protected areas added to the 62 existing protected areas. The mean candidate network area was 3138.5 km$^2$ (SD = 24.6 km$^2$), again slightly larger than the target of 3080 km$^2$. Network ecological representativeness scores ranged from 2.76 to 7.63 (mean = 4.71, SD = 0.64) (Fig 3). Network functional connectivity ranged from 4.21e$^{+05}$ to 4.79e$^{+05}$ (mean = 4.38e$^{+05}$, SD = 9.42e$^{+03}$) under current conditions (Fig 3A), and from 4.41e$^{+05}$ to 5.00e$^{+05}$ (mean = 4.58e$^{+05}$, SD = 9.51e$^{+03}$) under mean future conditions (Fig 3B). There was a negative relationship between ecological representativeness and functional connectivity, under both current (slope = -3922, p < 0.001, Fig 3A) and mean future conditions (slope = -3779, p < 0.001, Fig 3B).

The Ministry's proposed network (Fig 1) had an ecological representativeness score of 4.65 (Fig 3A), which was slightly lower than our sample mean. Its current and future functional connectivity were 4.59e$^{+05}$ and 4.76e$^{+05}$, respectively, both above our sample means (Fig 3A and 3B). However, our methodology yields many candidate networks that surpass the Ministry's proposed design in both representativeness and functional connectivity (Fig 3A and 3B). We conclude that the Ministry's design is suboptimal with respect to these three indicators of network effectiveness.

### Priority conservation areas

The quantile Q = 0.925 (92.5%-ile) yielded a subsample of 501 candidate protected areas scoring above the corresponding sample quantiles of all three indicators simultaneously (Fig 3). The ecological representativeness quantile was 5.68. The quantiles for current and future functional connectivity were 4.53 x 10$^5$ and 4.73 x 10$^5$, respectively. All 501 selected designs had higher ecological representativeness scores than did the Ministry's. Most of these 501 designs did not exceed the Ministry's functional connectivity scores, but some did (Fig 3): 78/501 designs had higher functional connectivity under both current and future conditions. That is to say, we identified some designs for the given tradeoff that outperformed the Ministry's design with respect to all three indicators.

All but one of the 690 candidate protected areas were included in at least one of the 501 selected networks. Selection frequencies were highly skewed (Fig 4). At the inflection point (32,35) on the graph, 32/689 candidate protected areas were included in at least 35/501 selected networks (Fig 4). These 32 priority candidate protected areas were fairly widely distributed over the study region (Fig 5), with some spatial clustering in the southwest of the Gaspésie National Park, and in the areas adjacent to the extreme west of the park (Fig 1), which includes the important high elevation caribou breeding habitats. The spatial distributions of high priority catchments under alternate conservation objectives were markedly different (S3 Material). In designs emphasizing ecological representation, priority areas were more widely distributed, but mostly in the south of the Gaspésie National Park, and none were adjacent to the Park (S3.1 Fig in S3 Material). In designs emphasizing only functional connectivity for caribou, all priority areas were adjacent to the Park (S3.2 Fig in S3 Material).

### Discussion

We present a methodology to define an effective protected areas network based on the tradeoff between ecological representativeness and functional connectivity over time, including the potential impacts of climate change. Our methodology yields a proposed subset of protected areas that may be close to ideal from an environmental or conservation point of view, but which does not fully respect all constraints of the use of public lands. For example, the Gaspésie region is highly disturbed by human activities. Considering all the social and economic constraints would have reduced too much the area for potential new protected area

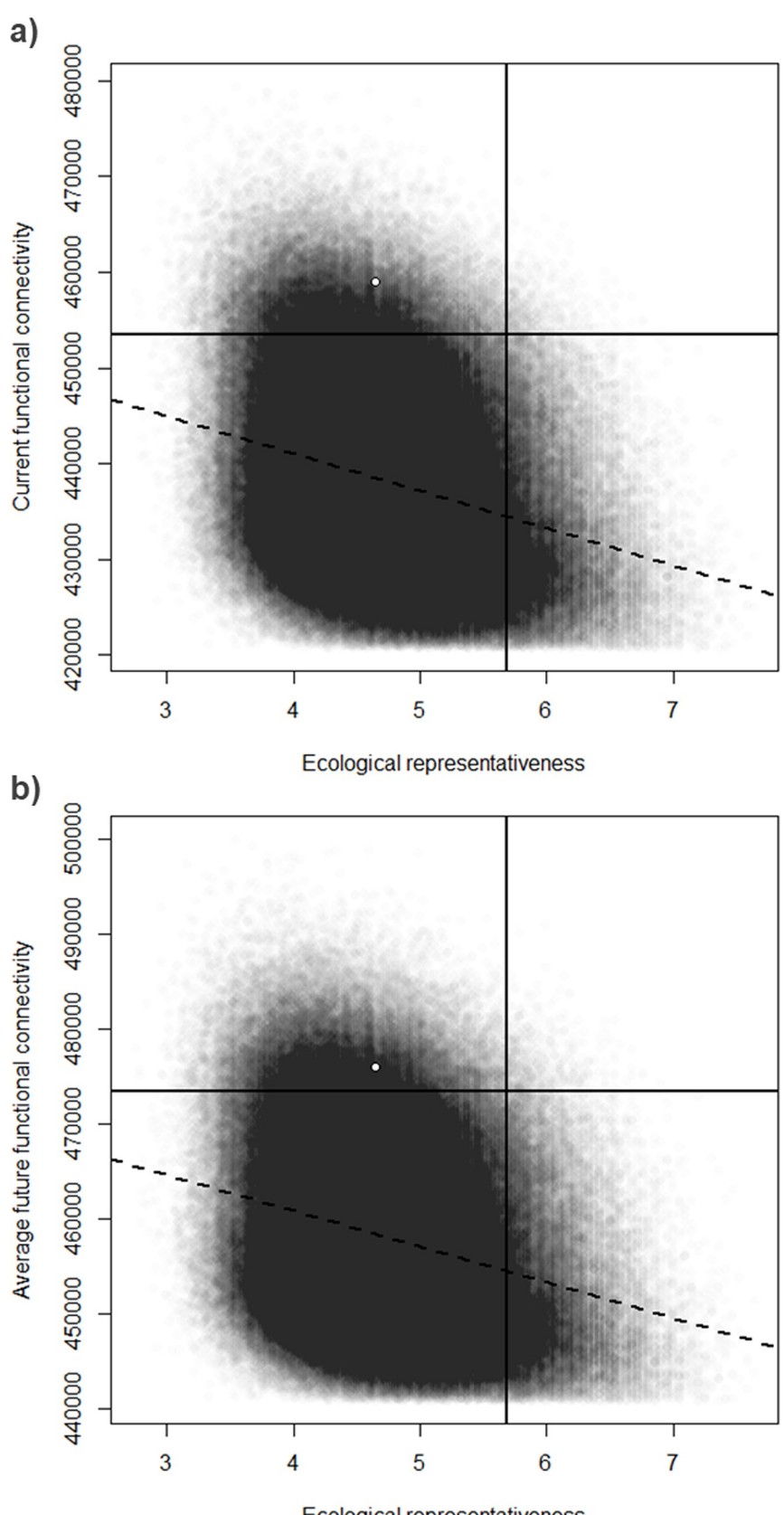

**Fig 3.** Scatter plots of ecological representativeness against functional connectivity defined for (a) the current time period and (b) the future average conditions for the 500,000 created networks. The solid lines represent the feature values at quantile Q = 0.925 (5.68 for ecological representativeness, $4.53e^{+05}$ and $4.73e^{+05}$ for current and future functional connectivity, respectively). The dashed lines represent fitted linear regression models of ecological representativeness against the functional connectivity measure plotted. The white dot represents the network proposed by the Québec government.

implementations, giving little space for designing different network scenarios or exploring the limits of what is possible. However, this methodology could easily include more features in the choice of the best network scenarios, and the proposed top protected areas would then be represented as the best tradeoff between all selected features. It would be possible, given data availability, to add constraints in the choice of the networks with, for example, the ecological representativeness of the future landscape under climate change, the functional connectivity of several species important for the ecosystem [68], the economic cost of excluding human activities from the proposed areas, the potential benefit with tourism if protected areas act as parks [69], or any of the other factors affecting the Ministry's design. Target features and constraints

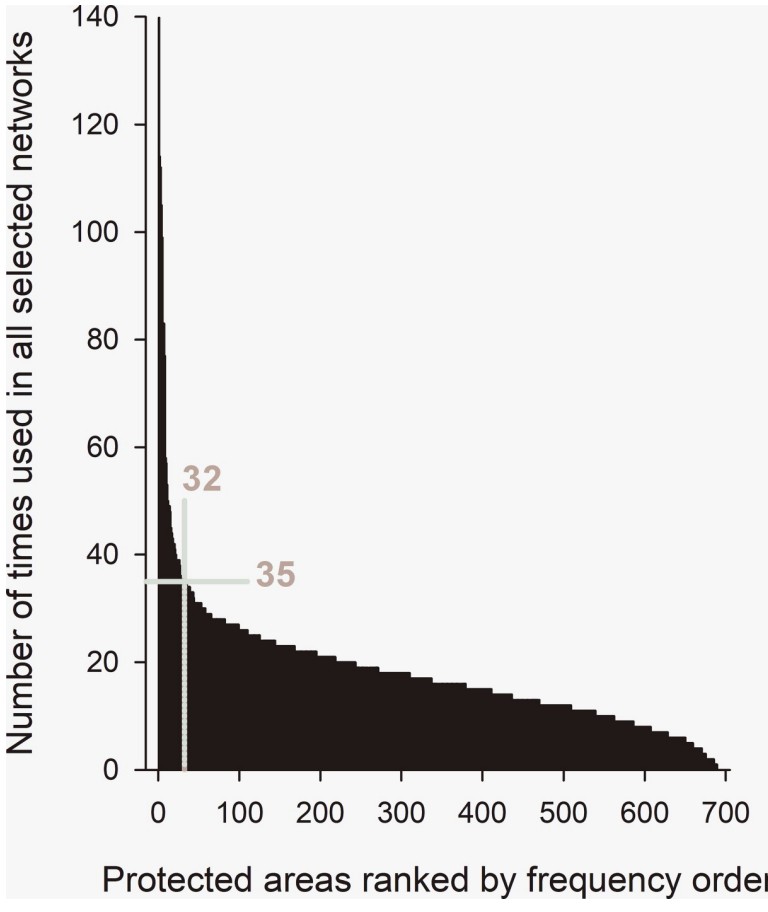

**Fig 4. A subsample of 501 candidate protected areas networks selected from near one point on the tradeoff curve (Fig 3).** These candidate networks included 689 of the 690 candidate protected areas (CPAs). The selection frequencies (*y*-axis) of these 689 CPAs are plotted in decreasing rank order (*x*-axis). Only 32/689 CPAs were included in more than 35/501 selected networks. This inflection point of the curve is indicated by the light grey lines perpendicular to the axes.

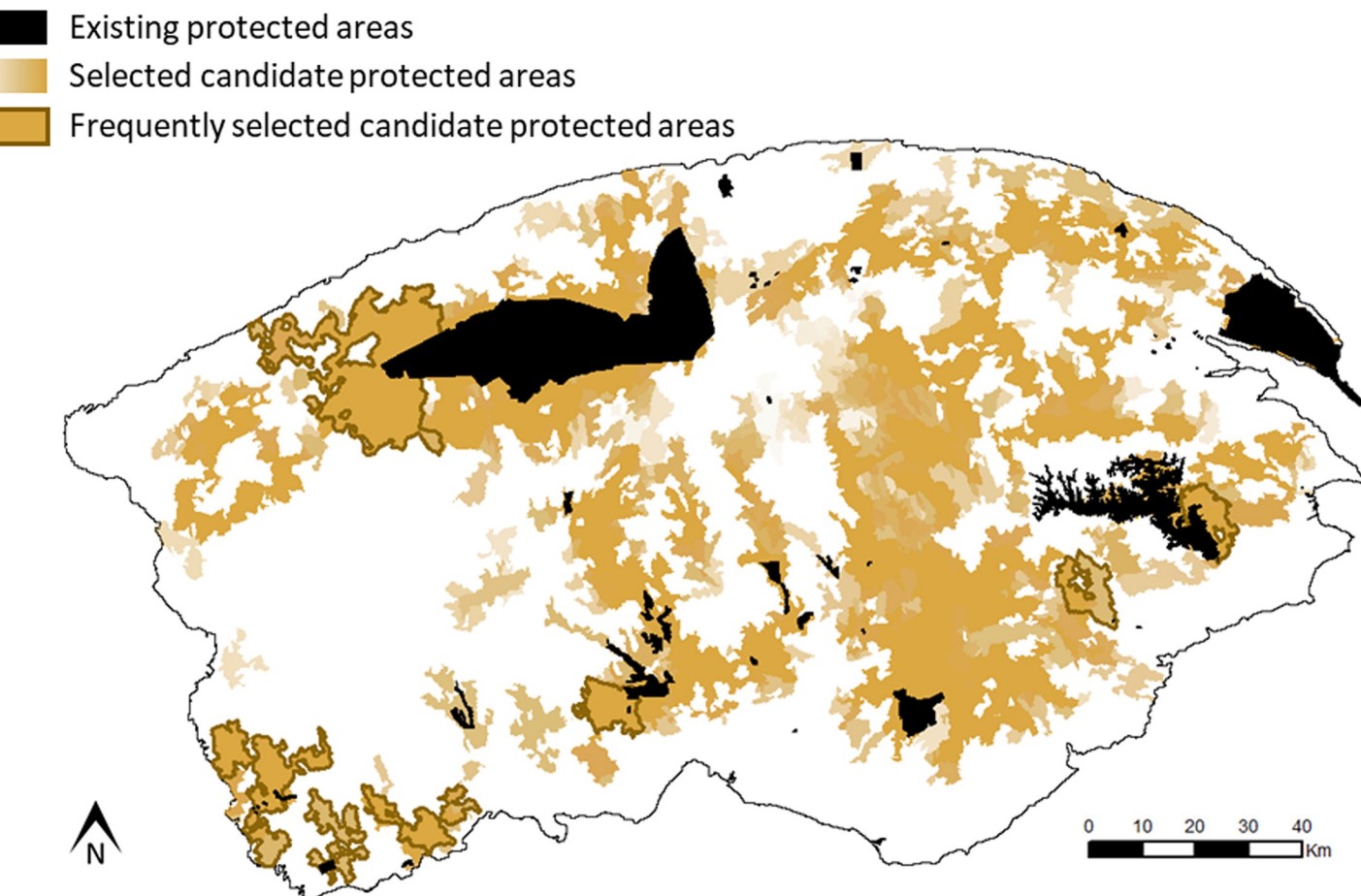

**Fig 5. The 689 Candidate Protected Areas (CPAs) included in the subsample of 501 near-Pareto-optimal networks are shown in orange scale, coloured by selection frequency, from low (light orange) to high (dark orange).** The 32 CPAs included in more than 35/501 networks (Fig 4) are outlined with a thick line. The existing protected areas are shown in black.

can be defined by local managers to help meet local biodiversity goals, and could easily be implemented in our method to improve the design of regional protected areas networks.

### Network ecological representativeness

Gap analyses are a common tool to quantify ecological representativeness of extant networks and to identify underrepresented features [e.g. 8, 70]. Here, we adapted an alternate two-stage approach that was recently developed in Canada [31, 32] to support systematic conservation planning in the boreal region. The first stage of that method used geospatial analysis tools to construct potential or candidate protected areas, which were then assembled into networks. A multivariate distribution matching methodology identified networks that minimized gaps. Gaps were quantified as dissimilarities, with respect to the distributions of a chosen set of environmental covariates, between the network and the rest of the study region.

### Network functional connectivity

Many studies have included connectivity in conservation planning using measures of distance or the cost of movement between protected areas [71] or applying graph and circuit theory

techniques [72, 73]. A novel feature of this study was that we derived network functional connectivity measures from movement simulations using an individual-based model [38, 61] that reproduced known characteristics of the focal population within the study region. The model accounted for process-based complex movement behaviors (e.g., seasonal site fidelity, foray loop movement) that would be difficult to represent in static habitat-based models. Being process-based, the models reflect behavioral responses to environmental conditions and should be preferred to make predictions under future conditions compared to models based on only observed habitat where the suppositions underlying current empirical relationships may no longer hold [74].

## Accounting for climate change

It is more efficient to take future climate change impacts into account now instead of reacting to them only once changes occur [28]. To account for climate change, we defined the future functional connectivity criteria using simulated movements on hypothetical landscapes resulting from different climate change scenarios. Network effectiveness will likely depend on the realized landscape outcome which is presently unknown. Therefore, networks selected using an average functional connectivity over a wide range of climate change impacts would be suboptimal for any particular climate change scenario. However, they might perform reasonably well under a large range of possible future environmental conditions. The differences in movement predicted for the different climate change scenarios resulted from the assumptions made about the future environmental conditions. We used this relatively simple approach as the functional connectivity differences among the scenarios were not particularly large. This suggests that using coupled climate change and vegetation change dynamic models may not dramatically improve the choice of protected areas in this region. Furthermore, since no climate-sensitive model of vegetation dynamics was available for Gaspésie natural region, using the scenario approach with averaging of results seemed to be a reasonable compromise to account for uncertainty in future conditions. Our approach could readily be adapted to cases with much greater divergence among expected future conditions, at the price of increasing the dimensionality of the tradeoff surface. Management decisions in that case would ideally be based on weighting the relative costs of possible solutions and the probabilities associated with each alternate outcome.

## Case study

The lack of functional connectivity during the last 20 years between the different major mountain summits of the Gaspésie National Park has led to a division of the caribou population into two genetically distinct sub-populations and is now jeopardizing the persistence of this isolated population [75]. Consequently, using an approach that could identify key elements to preserve in order to maintain functional connectivity could benefit the conservation of an endangered caribou population, especially under a changing climate.

In the Gaspésie region, there is only one caribou population occurring primarily inside the Gaspésie National Park [37, 48]. Consequently, in our analyses, networks with high functional connectivity tended to include many protected areas adjacent to or near this park. On the other hand, to achieve high ecological representativeness, protected areas needed to be more or less evenly distributed over the entire region to capture habitat diversity. There was therefore a negative relationship between representation and connectivity in our case study, as others have found [e.g. 76]. It is, in general, not possible to simultaneously optimize for multiple design criteria. Our sampling-based design methodology allows us to approximate a multidimensional tradeoff surface. Given the management decision about the choice of tradeoff, we

can identify a large number of solutions that are close to this point. Our methodology showed how to design new protected areas for a regional network that satisfy multiple, divergent criteria to (nearly) the highest degree possible. Other tradeoffs between the two objectives of representation and caribou conservation could be defined. The two extremes (S3 Material), in which only one criterion is emphasised, produce very different spatial distributions of priority candidate protected areas. Our chosen tradeoff produced an intermediate result, indicating that new protected areas in the south of the study region are needed to improve ecological representation, and that an expansion of Gaspésie National Park would best improve functional connectivity for caribou. However, the designs considered in Fig 5 are not intended or represented as optimal for caribou conservation *per se*.

The protected areas network expansion proposed by the Québec government to achieve the 12% coverage target that we used as a comparison was different than the sample of candidate protected areas suggested from the selected best networks identified here (Figs 1 and 5). The Ministry's network achieved a lower ecological representativeness than our selected networks. However, they had to respect design criteria and constraints, like socio-economic issues or the inclusion of rare ecosystems that we did not consider. This could explain the suboptimal ecological representativeness achieved by their network. Regarding functional connectivity, their network seems well connected from the Atlantic-Gaspésie caribou point of view. The performance of their scenario is surprisingly quite good for this feature, considering it was not explicitly part of their design. The networks we selected may not achieve such a high functional connectivity because our specific choice of intactness criteria resulted in some areas being excluded from any of our network solutions. In particular, we excluded some non-intact areas close to the Gaspésie National Park that were included in the Ministry's design.

## Supporting information

**S1 Material. Construction of the potential future landscapes.**
(DOCX)

**S2 Material. Caribou movements in the current and future landscapes.**
(DOCX)

**S3 Material. Description of the optimal protected areas networks.**
(DOCX)

## Acknowledgments

We thank the Canadian BEACONs Project (http://www.beaconsproject.ca) for developing and sharing methods and software for protected areas design, and for providing the custom hydrological catchments. We also thank BEACONs members M. Houle, K Lisgo and P. Vernier for their help with the use of the BEACONs toolkit. We thank the Ministère de l'Environnement et de la Lutte aux Changements Climatiques du Québec (D. Boisjoly, F. Brassard and S. Benoit) and the Ministère des Forêts, de la Faune et des Parcs du Québec for the data and documents provided. We finally thank K. Malcolm and three anonymous reviewers for their constructive comments on an earlier version of this manuscript.

## Author Contributions

**Conceptualization:** Steven G. Cumming, Eliot J. B. McIntire.

**Data curation:** Martin-Hugues St-Laurent.

**Formal analysis:** Sarah Bauduin, Eliot J. B. McIntire.

**Funding acquisition:** Steven G. Cumming, Martin-Hugues St-Laurent, Eliot J. B. McIntire.

**Methodology:** Sarah Bauduin, Steven G. Cumming, Eliot J. B. McIntire.

**Project administration:** Steven G. Cumming.

**Resources:** Eliot J. B. McIntire.

**Supervision:** Steven G. Cumming, Martin-Hugues St-Laurent, Eliot J. B. McIntire.

**Writing – original draft:** Sarah Bauduin.

**Writing – review & editing:** Steven G. Cumming, Martin-Hugues St-Laurent, Eliot J. B. McIntire.

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
