## [Decision Letter · Decision Letter 0]

19 Mar 2020

PONE-D-20-04608

Coping with climate change when designing networks of protected areas by accounting for functional connectivity: a case-study with caribou

PLOS ONE

Dear Dr. St-Laurent,

Thank you for submitting your manuscript to PLOS ONE. After careful consideration, we feel that it has merit but does not fully meet PLOS ONE’s publication criteria as it currently stands. Therefore, we invite you to submit a revised version of the manuscript that addresses the points raised during the review process.

The manuscript is sound and would make a nice contribution to conservation planning field. Both reviewers are positive and made extensive comments aiming at improving the papers, especially methods. I would like to see many comments included in the paper, and when this is not possible please explain why. 

We would appreciate receiving your revised manuscript by May 03 2020 11:59PM. To enhance the reproducibility of your results, we recommend that if applicable you deposit your laboratory protocols in protocols.io, where a protocol can be assigned its own identifier (DOI) such that it can be cited independently in the future. For instructions see: http://journals.plos.org/plosone/s/submission-guidelines#loc-laboratory-protocols

We look forward to receiving your revised manuscript.

Kind regards,

Laurentiu Rozylowicz, Ph.D.

Academic Editor

PLOS ONE

Journal Requirements:

3. We note that Figures 1 and 4 in your submission contain map images which may be copyrighted. All PLOS content is published under the Creative Commons Attribution License (CC BY 4.0), which means that the manuscript, images, and Supporting Information files will be freely available online, and any third party is permitted to access, download, copy, distribute, and use these materials in any way, even commercially, with proper attribution. For these reasons, we cannot publish previously copyrighted maps or satellite images created using proprietary data, such as Google software (Google Maps, Street View, and Earth). For more information, see our copyright guidelines: http://journals.plos.org/plosone/s/licenses-and-copyright.

1.    You may seek permission from the original copyright holder of Figures 1 and 4 to publish the content specifically under the CC BY 4.0 license. 

Reviewers' comments:

Reviewer's Responses to Questions

**Comments to the Author**

1. Is the manuscript technically sound, and do the data support the conclusions?

Reviewer #1: Partly

Reviewer #2: Partly

2. Has the statistical analysis been performed appropriately and rigorously? 

Reviewer #1: No

Reviewer #2: Yes

3. Have the authors made all data underlying the findings in their manuscript fully available?

Reviewer #1: No

Reviewer #2: No

4. Is the manuscript presented in an intelligible fashion and written in standard English?

Reviewer #1: Yes

Reviewer #2: Yes

5. Review Comments to the Author

Reviewer #1: Thank you for the opportunity to review this manuscript. While this is an interesting spatially explicit study on a topic of high conservation relevance and immediate urgency, I have several concerns. My main concern with this study is that it places considerable emphasis on ecological representativeness, which is basically the landscape variability in the study area. Whereas the stated goal (e.g., right from the title) is to help devise an optimal protected area network for endangered caribou specifically.

I understand the desire to represent the broader landscape in the protected area network. I also see the appeal of providing a spatially informed means to enable trade-offs between ecological representativeness and functional connectivity for caribou. I think this is a noble scope. Overall however I see this approach as lacking comprehensive consideration of caribou population needs, resulting in an analytical framework and reporting that do not adequately address the life history challenges of caribou in the study population. Perhaps the Individual Based Model (IBM) encapsulates a wealth of ecologically relevant information, but that needs to be detailed herein. In particular, what were the set of rules set for the IBM? Where do food availability and predation risk fit in here? Are these encapsulated in the functional connectivity component?

The authors must provide details on what caribou in their study populations actually prefer. I suspect that caribou like and do better in specific landscape features that do not reflect landscape diversity (representativeness). In fact they reflect quite the opposite, e.g. tundra or old-growth forest. There is therefore a discrepancy between ecological representativeness as quantified in this study and representativeness for caribou. The term "functional connectivity" used by the authors in relation to caribou does not imply consideration of core areas, breeding areas, essential foraging and resting habitat, refugia from predation etc. It simply pertains to movements between resource patches. I would like to see an emphasis on the resources that the caribou require and a detailed explanation of how the IBM via its rule set leads to the emergence of resource patch distributions.

Overall, my sense is that the paper is trying to come up with a protected area network for an endangered species; but not necessarily incorporating either best knowledge or a straightforward approach to facilitate desired landscape planning and conservation outcomes for this species. The attempt to include ecological representativeness of the broader landscape is a heavy burden that does a disservice to the protected area network outputs. I ponder on how this can be an effective strategy for an endangered caribou population with ~70 individuals total fragmented into 3 subpopulations.

I suggest a much more aggressive and caribou-tailored conservation strategy, including a protected area design framework tailored more specifically - exclusively - to the caribou population. This would maximize protected area representativeness for caribou, and would also eliminate the emphasis on the rather vague notion of broader ecological representativeness. Ecological representativeness as quantified with the 4 attributes the authors considered (elevation, surficial deposit, drainage class and potential vegetation type), has questionable conservation relevance. It also distracts from the goal of optimal protected area design for an endangered species. The 4 attributes, with the exception of vegetation type and perhaps elevation, are coarse scale and it is unclear how related they are, if at all, to biological diversity, richness and abundance of the various taxa in the study region. Furthermore, elevation does not seem an appropriate attribute as applied here, because the ecological representativeness model defines headwaters as catchment seeds, thereby biasing modelling outputs to high elevation (which do no not necessarily represent the broader landscape).

I have included additional comments in the attached .pdf.

Reviewer #2: Please see attached comments.

......................................................................................................................................................................................

6. PLOS authors have the option to publish the peer review history of their article (what does this mean?). If published, this will include your full peer review and any attached files.

Reviewer #1: No

Reviewer #2: No

---

## [Author Response · Author response to Decision Letter 0]

15 Jun 2020

Coping with climate change when designing networks of protected areas by accounting for functional connectivity: a case-study with caribou

(MS # PONE-D-20-04608)

Responses to reviewers’ comments

Please find below our responses to the reviewer’s comments; the line numbers in our responses refer to the revised version (clean copy) of our manuscript.

Please note that we have also uploaded a version of our revised manuscript with all modifications highlighted in “Track Change”.

JOURNAL REQUIREMENTS (WHEN SUBMITTING YOUR REVISION, WE NEED YOU TO ADDRESS THESE ADDITIONAL REQUIREMENTS)

REQUIREMENTS #1. Please ensure that your manuscript meets PLOS ONE's style requirements, including those for file naming. The PLOS ONE style templates can be found at http://www.plosone.org/attachments/PLOSOne_formatting_sample_main_body.pdf

and http://www.plosone.org/attachments/PLOSOne_formatting_sample_title_authors_affiliations.pdf

OUR RESPONSE: We formatted the title, authors and affiliations following PLOS guidelines.

REQUIREMENTS #2. We note that you have indicated that data from this study are available upon request. PLOS only allows data to be available upon request if there are legal or ethical restrictions on sharing data publicly. For more information on unacceptable data access restrictions, please see http://journals.plos.org/plosone/s/data-availability#loc-unacceptable-data-access-restrictions

OUR RESPONSE: We uploaded the relevant data we used in our calculation in a DRYAD repository (https://doi.org/10.5061/dryad.612jm641c). However, as the Atlantic-Gaspésie caribou population is currently endangered under the Canadian Species at Risk Act, we decided not to upload the telemetry location of the caribou in order to protect them from being tracked, disturbed, displaced or even poached by people. We hope you’ll understand that decision, but as the vice-president of the Atlantic-Gaspésie caribou recovery team, I cannot accept to increase the level of risk on those animals, sorry. However, the data can be made available upon request by qualified researchers if they contact Prof. Martin-Hugues St-Laurent at Université du Québec à Rimouski (martin-hugues_st-laurent@uqar.ca). Being co-owner of the data (along with the Québec Ministry of Forests, Wildlife and Parks), I’ll then have to transfer the request to the authorities of the QMFWP to obtain their permission.

IN YOUR REVISED COVER LETTER, PLEASE ADDRESS THE FOLLOWING PROMPTS

REQUIREMENTS #3. If there are ethical or legal restrictions on sharing a de-identified data set, please explain them in detail (e.g., data contain potentially sensitive information, data are owned by a third-party organization, etc.) and who has imposed them (e.g., an ethics committee). Please also provide contact information for a data access committee, ethics committee, or other institutional body to which data requests may be sent. If there are no restrictions, please upload the minimal anonymized data set necessary to replicate your study findings as either Supporting Information files or to a stable, public repository and provide us with the relevant URLs, DOIs, or accession numbers. For a list of acceptable repositories, please see http://journals.plos.org/plosone/s/data-availability#loc-recommended-repositories. We will update your Data Availability statement on your behalf to reflect the information you provide. 

OUR RESPONSE: We detailed our decision regarding these points in the cover letter.

REQUIREMENTS #4. We note that Figures 1 and 4 in your submission contain map images which may be copyrighted. All PLOS content is published under the Creative Commons Attribution License (CC BY 4.0), which means that the manuscript, images, and Supporting Information files will be freely available online, and any third party is permitted to access, download, copy, distribute, and use these materials in any way, even commercially, with proper attribution. For these reasons, we cannot publish previously copyrighted maps or satellite images created using proprietary data, such as Google software (Google Maps, Street View, and Earth). For more information, see our copyright guidelines: http://journals.plos.org/plosone/s/licenses-and-copyright. We require you to either (1) present written permission from the copyright holder to publish these figures specifically under the CC BY 4.0 license, or (2) remove the figures from your submission:

 1. You may seek permission from the original copyright holder of Figures 1 and 4 to publish the content specifically under the CC BY 4.0 license. We recommend that you contact the original copyright holder with the Content Permission Form (http://journals.plos.org/plosone/s/file?id=7c09/content-permission-form.pdf) and the following text:

 Please upload the completed Content Permission Form or other proof of granted permissions as an "Other" file with your submission. In the figure caption of the copyrighted figure, please include the following text: “Reprinted from [ref] under a CC BY license, with permission from [name of publisher], original copyright [original copyright year].”

OUR RESPONSE: The shapes we used came from the public domain, so no permission was required.

REQUIREMENTS #5. Please include captions for your Supporting Information files at the end of your manuscript, and update any in-text citations to match accordingly. Please see our Supporting Information guidelines for more information: http://journals.plos.org/plosone/s/supporting-information.

OUR RESPONSE: We have included an updated Supporting Information file, with captions just below the figures, i.e., not in a separate file as per instructions. 

 

REVIEWER #1

COMMENT #1. Thank you for the opportunity to review this manuscript. While this is an interesting spatially explicit study on a topic of high conservation relevance and immediate urgency, I have several concerns. My main concern with this study is that it places considerable emphasis on ecological representativeness, which is basically the landscape variability in the study area. Whereas the stated goal (e.g., right from the title) is to help devise an optimal protected area network for endangered caribou specifically.

OUR RESPONSE: Our goal was not to “devise an optimal network for caribou explicitly”. We but to incorporate aspects of caribou movement into a relatively standard protected areas design methodology based on ecological representation, intactness and hydrological connectivity. The ideas of optimality arise only in the evaluation of trade-offs between an index of representation and of functional connectivity for caribou, as explained in the Methods. We added an explicit statement at lines 370-382. We may have failed to clearly explain our objective, but we cannot identify in the text the source of misunderstanding. We hope the revised, abbreviated title does not mislead in this respect. 

COMMENT #2. I understand the desire to represent the broader landscape in the protected area network. I also see the appeal of providing a spatially informed means to enable trade-offs between ecological representativeness and functional connectivity for caribou. I think this is a noble scope. Overall however I see this approach as lacking comprehensive consideration of caribou population needs, resulting in an analytical framework and reporting that do not adequately address the life history challenges of caribou in the study population. Perhaps the Individual Based Model (IBM) encapsulates a wealth of ecologically relevant information, but that needs to be detailed herein. In particular, what were the set of rules set for the IBM? Where do food availability and predation risk fit in here? Are these encapsulated in the functional connectivity component?

OUR RESPONSE: The IBM includes many details of the behaviour and ecology of this caribou population, which we now briefly describe at lines 267-308 in the “Network functional connectivity” section. The model simulates caribou movement according to some elements of the landscape which account for both caribou resources/habitats/barriers/etc. as well as their predator’s habitats. These are predicted from a map of landcover, disturbance history and transportation networks using a Resource Selection Function model developed for this population, as cited in the revised version of the manuscript. 

COMMENT #3. The authors must provide details on what caribou in their study populations actually prefer. I suspect that caribou like and do better in specific landscape features that do not reflect landscape diversity (representativeness). In fact they reflect quite the opposite, e.g. tundra or old-growth forest. There is therefore a discrepancy between ecological representativeness as quantified in this study and representativeness for caribou. The term "functional connectivity" used by the authors in relation to caribou does not imply consideration of core areas, breeding areas, essential foraging and resting habitat, refugia from predation etc. It simply pertains to movements between resource patches. I would like to see an emphasis on the resources that the caribou require and a detailed explanation of how the IBM via its rule set leads to the emergence of resource patch distributions.

OUR RESPONSE: We completely agree there is a discrepancy between landscape diversity and habitat preferences for the caribou. This is visible in Figures 2 (negative relationships between caribou functional connectivity and ecological representativeness). That is why the best solution for protected areas networks are trade-off between these components as they cannot be both maximized. The entire of this study was to address this discrepancy by augmenting a conventional representation-based design with species-specific movement behaviour; we tried to explain this in the Introduction. As stated in our response to Comment #2, the functional connectivity was defined using the IBM which is based on habitat selection for several landscape elements: core areas = habitat of high quality; mating areas; and foraging and resting habitats. The functional connectivity is derived from simulated movements over the landscape where spatially and temporally varying movement behaviour integrates all these elements. A description of the rules governing the IBM which include all these elements was added in the manuscript in the “Network functional connectivity” section (lines 267-308).

COMMENT #4. Overall, my sense is that the paper is trying to come up with a protected area network for an endangered species; but not necessarily incorporating either best knowledge or a straightforward approach to facilitate desired landscape planning and conservation outcomes for this species. The attempt to include ecological representativeness of the broader landscape is a heavy burden that does a disservice to the protected area network outputs. I ponder on how this can be an effective strategy for an endangered caribou population with ~70 individuals total fragmented into 3 subpopulations

OUR RESPONSE: As we agreed above in responding to Comment #3, and as stated in the Introduction (see the lines 109-119 and 130-132) ecological representativeness and caribou “needs” are not the same, hence their negative relationship (Figures 2) and the need for a trade-off between them to obtain protected areas networks providing a range of levels of both characteristics.

The near-optimal networks presented in Figure 4 include many areas around the Gaspésie National Park to favor caribou persistence, as well as more distant areas across Gaspésie to account for ecological representativeness.

As explained in Introduction, this study attempts to add caribou considerations to an actual representation-based design to expand an existing protected areas network (see lines 128-130). Our aim was to show how such designs could be enhanced for caribou by including aspects of their space use and movement behavior. As stated above, we do not represent this work as an optimal conservation plan for this species or population (see lines 454-455).

COMMENT #5. I suggest a much more aggressive and caribou-tailored conservation strategy, including a protected area design framework tailored more specifically - exclusively - to the caribou population. This would maximize protected area representativeness for caribou, and would also eliminate the emphasis on the rather vague notion of broader ecological representativeness. Ecological representativeness as quantified with the 4 attributes the authors considered (elevation, surficial deposit, drainage class and potential vegetation type), has questionable conservation relevance. It also distracts from the goal of optimal protected area design for an endangered species. The 4 attributes, with the exception of vegetation type and perhaps elevation, are coarse scale and it is unclear how related they are, if at all, to biological diversity, richness and abundance of the various taxa in the study region. Furthermore, elevation does not seem an appropriate attribute as applied here, because the ecological representativeness model defines headwaters as catchment seeds, thereby biasing modelling outputs to high elevation (which do no not necessarily represent the broader landscape).

OUR RESPONSE: The goal of this study was to propose different protected areas networks that could be implemented in real life by the Government of Quebec. When implementing new protected areas in Gaspésie, the government’s goal is not only to protect the caribou population but also to protect the “common” elements of regional biodiversity, hence our analysis using other elements (i.e., ecological representativeness). If the goal would have been focused only on the caribou population, an extension of the Gaspésie National Park where the caribou are located would have been the answer.

The selection of biodiversity surrogates in representation analysis is a vast topic but the use of surrogates is very widespread. Our specific choices of surficial deposit, drainage class and potential vegetation were made in conformance with the Provincial government’s design process which we are evaluating and augmenting, as explained in section Network ecological representativeness. To these we added elevation, precisely to counter any elevational bias in catchments selection due to rooting the benchmark construction at headwater catchments. We don’t think it requires pointing out in the text that general method we present does not depend on the choice of surrogates. 

ADDITIONAL COMMENTS IN THE EDITED PDF DOCUMENT

COMMENT #6 - Title. Please shorten the title, in its current form it is a bit of a mouthful. Titles of up to 15 words are generally preferable.

OUR RESPONSE: The title has been abbreviated to 12 words, as follows: “Protected areas networks, functional connectivity and climate change: a caribou case-study.”

COMMENT #7 - Authors. Order of authors is different here than at the bottom of pg 1 of this document. Please adjust for consistency.

OUR RESPONSE: This was an error when submitting the manuscript. We corrected that in the resubmitted version, thanks for noticing.

COMMENT #8 – L27. “…and likely help…” change to “and when designed adequately likely help…”.

OUR RESPONSE: Done.

COMMENT #9 – Lines 28-30. Statement is a bit too generic and needs changing. In fact, some protected areas are fairly small in size and/or designated for focal species conservation rather than for biodiversity protection. E.g., an endangered plant species, or a plant at the limit of its distribution range.

OUR RESPONSE: We adapted the sentence.

COMMENT #10 – Line 50. Order keywords alphabetically.

OUR RESPONSE: Done.

COMMENT #11 – Lines 92-93. “Therefore, the ability of a protected areas network to sustain functional connectivity under is another dimension of effectiveness”. What “under” means here? Needed?

OUR RESPONSE: The intended reading was “under climate change”. This has been corrected.

COMMENT #12 – Lines 98-100. Please add more background information on current state of knowledge and methodologies for protected area designation. Illustrate these with examples from varied taxa supported by references to literature.

OUR RESPONSE: We regret that it has not been possible to conduct such a review. We recognize that it could have been very interesting but it would have greatly increased the length of our manuscript. We thus decided that this suggestion could be left unanswered.

 

COMMENT #13 – Lines 115-117. Insert references at the end of this sentence.

OUR RESPONSE: We have added a subject-matter relevant reference for Pareto optimality at lines 108-112, with reference to the following article: 

Kennedy, M.C., Ford, E.D., Singleton, P., Finney, M. and Agee, J.K., 2008. Informed multi‐objective decision‐making in environmental management using Pareto optimality. Journal of Applied Ecology, 45(1), pp.181-192.

COMMENT #14 – Line 185. Why were highly disturbed types, e.g. "urban areas" and "agriculture", assigned the same weights as some of the other disturbances? This seems likes a major flaw. Forestry activities, roads and trails can trigger temporal avoidance but animals can still use these areas; whereas they would not be able to use urban areas and presumably avoid agricultural areas also.

OUR RESPONSE: The intactness of the catchment was defined using these 6 disturbances and these elements were included to build intact protected areas to conserve biodiversity as such, not only caribou. Giving different weights to the different disturbances seemed subjective as some disturbances can have a large impact on some species but a minor impact on other species. For example, agricultural fields have a large impact on caribou but less impact on meso-carnivores. Even urban areas can be less avoided by some species (e.g., birds, insects) relative to either heavily forested areas or highways and their surroundings. At worst, our criteria could be regarded as too conservative. We added a sentence in the text to clarify this point (see lines 213-221 and the last paragraph of the discussion). Regarding the application of our method to the study area, please note that agricultural fields and urban areas are very rare and all distributed along the coast, not in the middle of the peninsula where caribou can be found. Finally, please note that the caribou IBM effectively uses empirical weights for the effect of different feature classes via the RSF model (see Bauduin et al. 2016; Ecological Modelling).

COMMENT #15 – Lines 192-195. Could you instead simply clip the rasters at the end? This way you could still have a model output for private land, in case private land acquisition becomes feasible for protected area expansion.

OUR RESPONSE: This action needed to be done at this initial stage so that catchments on private lands, which are not available for inclusion in the Province’s proposed expansion, are not included in the construction of candidate protected areas.

 

COMMENT #16 – Line 198. Ok, but I thought your model organism was caribou. How are caribou responding to headwater catchments? Does empirical data show that they utilize these headwaters, and if so, to what intensity compared to downstream areas in the catchments?

OUR RESPONSE: The construction of the candidate protected areas is done to protect biodiversity in its most “untouched” way and is not focused on our case study and the caribou. Using headwater catchment is done to ensure terrestrial and hydrological connectivity and conserve biodiversity features (like fish and aquatic invertebrates) that may depend on it. 

This specific feature of the benchmark design is not aimed at terrestrial organisms and is unrelated to our case study. We added a few sentences to highlight this point and avoid confusion (lines 219-221).

Again we stress, any benefit to caribou is an emergent property of networks, not an attribute of individual catchments. Similarly, network characteristics address other goals than caribou.

COMMENT #17 – Lines 207-209. What does a protected area of this size contribute to the conservation of a wide-ranging species such as caribou in your study system? Are caribou migratory in your system and have you incorporated this life history characteristic in your simulation framework? What are the main causes of caribou decline in your area and how does the intactness metric relate/account for them? E.g. do areas with higher intactness have better nutritional availability and lower risk of calf mortality through predation? Use references to justify.

OUR RESPONSE: Connectivity and ecological representation are characteristics of the network, not of any specific component. The presence of small protected areas as such within a network does not disqualify it. The definition of the protected areas is the first step to create an effective network of protected areas that may be implemented in real life. It should not be defined for the caribou only. We added a few sentences to highlight this point and avoid confusion (lines 218-221).

Moreover, the caribou population in Gaspésie is not wide-ranging in their current state. These animals make relatively short-distance seasonal migrations from the top of the mountains to adjacent or nearby forested areas below the tundra (Mosnier et al. 2003). 

COMMENT #18 – Lines 232-233. Who decides which resolution is "adequate"? What is that resolution?

OUR RESPONSE: We meant that the resolution was high enough compared to the size of the catchment to be meaningful at the catchment level. We removed this info about resolution to avoid confusion and because it is not relevant.

COMMENT #19 – Lines 233-234. What is a "standard spatial data operation"? Describe briefly what you did.

OUR RESPONSE: We added some text to spell it out (see lines 243-247).

COMMENT #20 – Lines 235 +(and the following). Use past tense throughout.

OUR RESPONSE: Done.

COMMENT #21 – Line 269. …subpopulation. Add a S at subpopulation..

OUR RESPONSE: Done.

COMMENT #22 – Lines 271-272. Why four-year simulations and not annual? With such low population sizes, it seems that environmental factors e.g. inter-annual variability in winter severity could really impact the population.

OUR RESPONSE: The IBM simulating caribou movement includes an attraction of the mating area but only during mating season. To be able to capture the movement towards and away these areas, with just a few individuals, we needed to run the simulation longer than one year. The IBM model has many stochastic processed and to capture the whole range of movement possibility, the simulation needed to run a few years with so few individuals to adequately assay functional connectivity. We did not simulate caribou movement for more than 4 years because of computer capacities and considering that we reached similar responses after more than a few years.

COMMENT #23 – Line 272. I thought the spatial grain was 250 x 250 m?

OUR RESPONSE: The resolution 250 x 250m is used to define the intactness. The resolution of 1-ha applies to the IBM. The resolution of these two rasters are different but they are the best ones available for their respective use. Rasters are summarized at the catchment level for the intactness and at the network level for the caribou connectivity. There is no interaction nor conflict between the two resolutions.

COMMENT #24 – Lines 274-275. Does your model differentiate repeat movements within the same protected area, from movements in different protected areas? The former represents protected core area fidelity, whereas the latter addresses connectivity.

OUR RESPONSE: We did not look at this because the protected core area is the Gaspésie National Park and individuals rarely leave the park (see Mosnier et al. 2003; Morin 2018; but also based on the GPS telemetry data collected by MH St-Laurent during the last years and used in Lesmerises et al. 2017, 2018a,b and 2019). In this instance, as it turned out, our connectivity refers more the to the facilitation of seasonal movements rather than exchange of individuals among core areas.

COMMENT #25 – Lines 280. How was this assessed? Because of your large sample sizes in number of replicates, P-values associated with mean comparisons are not reliable. Did you use effect sizes?

OUR RESPONSE: No means tests were conducted. Every candidate protected area (CPA) was evaluated by repeated simulation of the IBM under each of the four climate scenarios. This resulted in four functional connectivity metrics per CPA. We evaluated overall function connectivity of a CPA by taking the mean of these four metrics. We slightly revised to clarify (see lines 312-318). 

COMMENT #26 – Lines 285-287. Unclear here, please provide more details.

OUR RESPONSE: We hope the revised text is easier to understand. It would be difficult to be more precise without introducing a bunch of notations. 

COMMENT #27 – Line 292. “inflexion” should be “inflection”.

OUR RESPONSE: Done.

COMMENT #28 – Lines 314-315. Replace “well below” by "which was slightly lower than" and remove “well” at line 318.

OUR RESPONSE: Done.

COMMENT #29 – Line 315. our sample mean was 4.71, correct? Not that different than 4.65.

OUR RESPONSE: It is true the value of 4.65 is not much smaller than the sample mean so we used your proposition as “slightly lower”.

COMMENT #30 – Line 434. Add reference to Fig. 1 too.

OUR RESPONSE: Done.

COMMENT #31 – Fig. 3 (caption). Provide better axes labels to reflect "protected area" and "protected area network".

OUR RESPONSE: Done (now Figure 4). The x-axis represents all the protected areas used in the selected networks, ranked in their order of selection in the network. The y-axis is their frequency.

REVIEWER #2

This manuscript proposes an approach to identifying protected area networks using simulation modeling that seek to balance multiple ecological criteria. The authors focus on ecological representativeness and functional connectivity as their metrics for optimization, with the former relating to broad landscape features and the latter focusing on an endangered species in Canada – the Atlantic-Gaspésie caribou population. They consider potential networks of protected areas under both current conditions and future conditions of climate change and timber management. The paper appears methodologically sound, though several of the descriptions should be clarified and supported with figures to make the approach clearer to readers. With the addressing of the issues below, I think this paper would make a worthy contribution to PLoS ONE.

COMMENT #32. The primary improvement needed for this paper is clarification of the methodology and support with additional figures. This paper really is focused around the details of its methods as it proposes a detailed methodology for conducting optimization. However, there are multiple places where methods are difficult to determine based on the information as currently organized, especially for those unfamiliar with the BEACONs approach. Given the spatial nature of the proposed simulation approach it is especially difficult to visualize the steps of the method from the description given. I had to read the methods multiple times to wrap my head around the steps the authors take to simulate candidate protected area networks. 

As best I can tell, the procedure that was used was to:

1. Identify a set of initial “seed” catchments

a. Defined here as all intact headwater catchments

2. For each seed catchment, add contiguous catchments that are hydrologically connected and meet minimum size and intactness criteria

a. Stop adding catchments when target size is achieved or no more catchments that meet the criteria are available to be added

b. Define the set of catchments associated with a given seed as a “candidate protected area” if its total area is greater than 31.2 sq km.

3. From the full set of candidate protected areas, generate “candidate protected area networks” by randomly selecting candidate protected areas and adding them to the existing protected area network until the total area exceeds 3080 sq km.

a. Repeat this step 500,000 times to generate many possible candidate protected area networks.

The paper would greatly benefit from clearly stating the above steps and from addition of one or two figures clarifying these steps. How protected area networks are generated is central to the methods and validity of the paper. I recommend the steps above be clearly indicated in the text and perhaps in a flowchart, with one or two figures displaying the corresponding spatial depictions:

All seed catchments across the study area, shown in relation to existing protected areas.

One candidate protected area, with each of the included catchments shown distinctly as outlines.

The full set of existing and candidate protected areas, displayed with transparency or some other way of indicating where overlap occurs.

A multi-panel plot showing how one candidate protected area network is compiled as an accumulation of the randomly selected candidate protected areas. Each step in the compilation does not need to be shown, but a representative selection of steps throughout the process should be displayed to give the reader an understanding of how networks were created. By including the total aggregate area of the protected area network at each displayed step, it will be easy for the reader to understand how the total area threshold was met and how overlap between candidate protected areas did not alter the candidate network area.

o This will greatly clarify L.213-222. The discussion of overlap and its consequences (or lack thereof) was confusing without some frame of reference on which to base the statements. Showing where multiple candidate protected areas overlap (e.g., by displaying them with transparency in the figure) may help explain what is meant here. If this is made clear in the figure, then L.213-222 could be removed or greatly simplified. As long as the methods are clear that where candidate protected areas overlap only the aggregate area was considered (as indicated in the associated figure), that should be sufficient for the reader to understand what was done and why.

The above figures would likely benefit from use of color. Since PLoS ONE is online-only this will not result in extra charges.

OUR RESPONSE: The reviewer’s description of the three stages or the procedure is correct.

To address the recommendations, we have developed a new Figure (Figure 2) which illustrates these steps with explanation and shows also the initial catchments with headwater catchments distinguished. We have also expanded the explanations somewhat throughout the Methods and developed a colour version of the former Figure 4 (now Figure 5). 

COMMENT #33. Another helpful figure would be to depict functional connectivity. This is one of the two metrics employed in paper and yet the reader is left without a clear depiction of how functional connectivity actually works on the landscape. This also is relevant to L.414-416 in the Discussion. The paper states that a lack of functional connectivity during the last 20 years has led to division into sub-populations and heightened risk of extinction for caribou. This makes me wonder whether functional connectivity was overestimated in the models as it currently is hindered on the landscape. Without seeing any functional connectivity plots it is not possible to know how reasonable the model results for current and future functional connectivity might be. If the final functional connectivity map is considered protected information, since the caribou are an endangered population, a single iteration of the simulation could be shown as an example. This would at least give a sense of the model results. With this should be an indication of the location of the three caribou subpopulations references in L.269, as this will help the reader interpret the reasonableness of the functional connectivity results.

OUR RESPONSE: We added the figures of the summed caribou movements on the current landscapes and as averaged over the four different future landscapes to S3 – Supporting Information, as this is not the focus of the study. The results also appear in Bauduin, S., McIntire, E.J.B., St-Laurent, M.H., Cumming, S.G., 2018. Compensatory conservation measures for an endangered caribou population under climate change. Sci. Rep. 8, 1–10. https://doi.org/10.1038/s41598-018-34822-9, as cited [55].

COMMENT #34. Modifications are also needed for Figure 4 to provide consistency and clarity with the results. Based on the text in L.474, a continuous color ramp of greyscale values was used, but only a single grey is shown in the figure legend. Also, as it currently stands the legend seems to indicate that the frequently selected candidate protected areas are shown in white, while in actuality these are the dark areas with a black outline. Changing this square to have a darker grey fill with a black outline would clarify this. Use of color, which does not cost extra in PLoS publications, would also be helpful here. Finally, the text refers to frequently selected candidate protected areas being adjacent to important caribou breeding habitats (L.337-338). Since these important breeding areas are referenced, they should be shown in Fig. 4 or Fig. 1.

OUR RESPONSE: We fixed the mistakes on the legend for Figure 4 (now Figure 5) and used color for better clarity, thanks for noticing. 

The important caribou breeding habitats are the high-elevation areas within Gaspésie National Park (see Mosnier et al. 2008), and are thus effectively shown in Figure 1. We have rephrased the text formerly at lines 365-367 to remove the ambiguity.

Other methodological clarifications are also needed to allow the reader to assess the validity of the proposed methods:

COMMENT #35. L.165-179. Prior to this point the reader has only been introduced to the existing protected areas and those proposed by MELCC, as shown in Fig. 1. It was not initially clear whether the approach to building protected areas networks outlined here would apply only to the proposed protected areas or to all lands in the study area. This should be clarified.

OUR RESPONSE: The approach is applied to the entire study area, but in this specific application is additive to an existing protected areas network, up to the area target. We added a new paragraph at the beginning of the Method (a section called “Overview”) to introduce the goal of the study and present an overview of the method, adding here the new Figure 2 presenting the workflow, as suggested in your Comment #32. 

COMMENT #36. L.183-184. Roads and trails are likely to be much smaller than a 250 x 250m raster pixel. Was any pixel overlapped by a road or trail considered to be a road or trail?

OUR RESPONSE: Yes, as stated on the original manuscript in the following paragraph. We have reorganised slightly to avoid confusion and make this easier to understand.

COMMENT #37. L.184-185. The six disturbance types were given equal weight. Is there a reason to think that forestry activity and power line rights-of-way are as disruptive as roads or urban areas? In the absence of studies on caribou selection and avoidance equal weighting might be a reasonable assumption, but some support should be given from the literature for why caribou would be expected to respond strongly to things like forestry, rights-of-way and agriculture.

OUR RESPONSE: The measure of intactness is not directed primarily at caribou. It is rather one of the design criteria for protected areas networks, possibly a conservative one in that. The empirical data for the effects of specific disturbance types on caribou behaviour are incorporated into the Resource Selection Function which partially drives the caribou movement model (more details provided at lines 301-314). These responses effect the functional connectivity, which is used to select among candidate networks, each of which satisfies the network design criteria of e.g. representation, intactness and hydrological connectivity. We do not think any additional text is needed at this point in the manuscript. See also our answer to Comment #4 above.

COMMENT #38. L.185. Please give an example or two of what would be part of the “other” category.

OUR RESPONSE: We cannot provide these details as data we were provided did not specify. As stated, most of the disturbances were forest harvesting operations of roads and trails. Forest inventory data from other parts of Canada would suggest that minor residual areas of disturbance would be things like borrow pits for road construction, or clearings for infrastructure like electrical transformer stations.

COMMENT #39. L.191-192. Is there any evidence that the median intactness outside of existing protected areas is functionally relevant for caribou movement? In other words, is having < 0.5% of a catchment intact really enough to allow for caribou movement? I have some doubts about this. Without a study of caribou movement with respect to intactness it will be difficult to determine what is possible, but at least it would be good to indicate why this is a reasonable assumption. Was any sort of sensitivity analysis conducted to see how this threshold affected the resulting networks?

OUR RESPONSE: See above our response to Comment #37.

COMMENT #40. L.205-206. I do not understand what this means. Please clarify. For example, what does “protected area-level intactness criteria” mean? Is the point that intactness was only assessed at the catchment level, not for the resulting protected area? If so, this is not clear; I had to read it many times to come to this possible interpretation.

OUR RESPONSE: In some applications, it is useful to have separate intactness criteria at the catchment and CPA level: for example, in building large areas in regions with significant localised disturbances, one can allow a small number of highly impacted catchments within a CPA so as to that allow construction of a large CPA that is otherwise mostly intact. However, we did not use this feature in the present study, so the catchment and CPA-level criteria are effectively the same. To simplify the exposition we removed the reference to this unused feature of the construction software.

COMMENT #41. L.233-234. “Catchment-level attributes were calculated using standard spatial data operations.” What does that mean? I do spatial analysis frequently, but I am uncertain what the authors consider “standard spatial data operations.” This needs clarification. If the point is simply to introduce the approaches of the following paragraph then delete this sentence, as it does not describe ‘standard measures.’ If it means something like calculating the mean or median across pixels, then state that instead.

OUR RESPONSE: See above our response to Comment #19 made by Reviewer #1.

COMMENT #42. L.260. Consider updating to state that habitat generation was “to represent the current landscape and four landscapes for 2080 under future timber management and different climate change scenarios.” As currently stated, it is unclear how the current conditions map would differ from the scenario with no climate change impacts (L. 268) because it seems that only climate scenarios were changed. Based on the supplementary materials, however, it seems clear that timber management would lead to changes between the CC0 and current conditions map, even in the absence of climate effects. This should be indicated in the main text so those who do not read the supplement can still follow the basic approach.

OUR RESPONSE: We added a clause to clarify this point in the revised manuscript (see lines 315-328).

COMMENT #43. L.257. For readers not familiar with the approach of Bauduin et al. 2016 it would be helpful to give a brief summary of what the model does. For example (I made this up based on the information given here, not having read Bauduin et al. 2016), “The model simulates movements of individual caribou at an XX timescale with movement probability driven by habitat attraction/repulsion and an individual’s internal state [maybe an extra detail or two needed here]. Model outputs report the number of visits per landscape cell.” Such statements would give the reader the context to understand what was done and evaluate its relevance to the representation of functional connectivity. This could also be integrated with the content of L.269-272.

OUR RESPONSE: We added a summary of the main features of the IBM (see lines 268-281).

COMMENT #44. No indication is given for what software were used to conduct the simulations, making replication difficult. This should be clarified. While the authors explain why the input caribou data cannot be shared due to endangered species considerations, it does seem feasible to share code or other materials that would allow others to apply their modeling approach.

OUR RESPONSE: The BEACONs tool chain consists of two substantial pieces of custom software: the program that constructs candidate protected areas from a catchment shapefile and ancillary GIS layers, and a system that constructs and ranks candidate networks from the “Builder” output. Neither has yet been published other than as MSc theses: the reviewer may wish to refer to www.beaconsproject.ca for some background material. BEACONS has recently secured some resources to rework these tools as an R package, which would make them public, but this will take some time, and is not the responsibility of our former graduate student Dr. Bauduin. The custom R software used in the present analysis has been provided as supplementary data in the DRYAD depository.

COMMENT #45. The methods for generating future landscapes under climate change and management laid out in the supplementary materials also need some additional clarification. My primary concern is about how uncertainty was accounted for in generation of future landscapes. As best I can tell, future landscapes were only generated once for each climate scenario. However, at multiple steps in the process random selection and application of probabilities was applied (e.g., mature fir mortality due to spruce budworm, selection of fir stands outside of protection areas and conversion to other classes). If this is the case, the effects in future landscapes could be an artifact of these random selections and probabilities. Notably, the treatment of alpine habitat differed from that of fir habitat, as the former used fixed buffer distances, indicating that alpine habitat was always treated the same. This could result in different levels of variability across habitat classes. What impact did the random selection and use of probabilities have on outcomes? This paper generally does a good job of exploring many iterations of simulations to reflect potential variability, but here it is not clear that this was done. Some sort of sensitivity analysis or discussion would be nice.

OUR RESPONSE: The reviewer is correct. Only one future landscape was developed per scenario. The randomisation was at the level of individual mapped forest polygons which numbered in the hundreds of thousands. There’s no reason to expect that a specific random sample would introduce bias. Similarly, while developing these future landscapes, there was no discernible impact on caribou or protected area conclusions whether one set of mature fir polygons was removed or another set of mature fir polygons. Another level of replicated simulations would be very time consuming, and of doubtful interest or impact. 

Other aspects of the Supporting Information in which clarity is needed:

COMMENT #46. Alpine tundra paragraph: What were the buffered areas replaced with? I assume the classes along their borders. What happened if there was a mix of mature and regenerating fir stands along the boundary of alpine tundra.

OUR RESPONSE: Recall (see Supporting Information S1): The three vegetation types of interest were the alpine tundra, the mature fir stands and the regenerating stands; also recall: these are defined in more detail in other, published papers that we cite. Any pixels that ceased to be alpine tundra were simply part of the remainder of the rule sets i.e., there was no difference in these pixels or other “non-tundra-affected” pixels. To summarize here, for non-protected areas, this would follow the forest harvesting and climate change rule sets: 1) if ministry of forests project an increase (or decrease) in mature fir in a zone (there were 5 zones in our study area), then took the aspatial ministry forecasted proportion of “increase in mature fir” and turned a random set of polygons into mature fir (or turned mature fir into ‘other’), 2) for climate effects, we then impose mortality on the mature fir stands according to the intensity of the climate effects (0% up to 50% for the most severe scenario). In the protected areas, we imposed the natural disturbance effects (increase in age of some fir polygons, and some mortality) and then the climate effects (as per above). Thus, there could be new mature fir or removal (and same for regenerating stands), but the fact that it was tundra before didn’t change our implementation of rules: these simply because polygons on the landscape that could have been turned into mature fir.

COMMENT #47. Mature fir stands and regenerating stands section/Impact of climate change section: Here for the first time there is mention of an “other” category. Above only three habitats were mentioned as being used to generate habitat quality maps: alpine tundra, mature fir, and regenerating stands. How many habitat types were there actually? This needs to be made clear at the outset of the description. Also, what was lumped into the “other” category? The descriptions further in the supplement make it seem like it could contain fir trees from 30-50 years old, but this likely is very different functionally than if it consists of urban areas, agricultural areas, etc.

OUR RESPONSE: We added a mention earlier of the “other” types, which could also help for you previous comment. Habitat types (young fir stands, urban areas, agricultural areas, etc.) were distinguished (or not) depending on whether they were distinct in the RSF model used as input to the IBM. Minor changes were made to the manuscript at lines 194-200 but also to the Supporting Information. The categories were initially detailed in Gaudry 2013 (MSc Thesis – Université du Québec à Rimouski) and also in Bauduin et al. 2016 (Ecological Modelling), 2018 (Scientific Reports). The “other” category gathered the following landcover types: deciduous, spruce and mixed stands (all age classes confounded) and < 50-year-old balsam fir stands, as these were shown to be significantly less used than available (see Ouellet et al., 1996 and Mosnier et al., 2003).

COMMENT #48. Mature fir stands and regenerating stands section/Impact of climate change section: If potential habitat was determined to be lost, why were mortality probabilities set at < 1? It seems odd that this would happen. I understand doing this when habitat quality decreases, but not if it is lost. Without suitable habitat why should any fir forest be expected to remain, especially ≥ 50% of the time?

OUR RESPONSE: By “loss of habitat” we mean “changes such that the habitat becomes unfavourable for the tree species presently dominant at the site”. This does not mean all individuals of the species immediately die. The time horizon we chose, 2080, is only 60 years from now. Even if the habitats are predicted to become unsuitable by then, it is not realistic to think existing canopies or regenerating trees of the species present will have all died. We account for partial morality of 0.01, 0.1 and 0.5 given the severity of climate change, but we do not expect more than 50% of mortality in a system where wildfire is nearly absent. 

COMMENT #49. Mature fir stands and regenerating stands section/Impact of disturbances inside protected areas section: Good job identifying the potential for fire and windthrow to change with climate alteration as well as why these factors were not considered further. I would like to see more of this in the paper.

OUR RESPONSE: These were simplifying assumptions given the fact that the impact of these two disturbances is expected to be small relative to spruce budworm outbreaks. However, we have no study to support this. Developing integrated climate sensitive models of vegetation dynamics, insect defoliation, fire and human activities are beyond the scope of this study; such models may one day exist to provide more reliable inputs to future studies. 

COMMENT #50. Mature fir stands and regenerating stands section/Impact of disturbances outside protected areas: What was done in the remaining 28% of forest outside of protected areas (if BFEC units cover 72% of the forest outside protected areas)? There is no description given for how habitat in these areas was treated.

OUR RESPONSE: The 28% of forest outside protected areas were not public forests, there were either private forests (included in the analyses with the same rules as the public ones) or non-forested areas. Thus the BFEC do not have any rights to plan and operate commercial timber harvesting on private lands without the authorization of the owners (citizens). 

COMMENT #51. Mature fir stands and regenerating stands section/Impact of disturbances outside protected areas: It would be great to have a little more detail about how forestry plans for harvest are expected to increase the amount of old forest. This seems a bit counterintuitive if “the landscape is managed for timber production” as is stated above. I’m sure there’s a rationale for it, but for readers unfamiliar with the Québec context it would be great to explain briefly (e.g., by focusing timber harvest on currently regenerating stands and allowing older stands to grow unharvested…or whatever they are doing).

OUR RESPONSE: This refers to silvicultural treatments implemented as part of ecosystem-based management strategies designed to generate old forest conditions relatively early in the development of regenerating harvested stands. Our scenarios respect the plans as given. 

COMMENT #52. Another area where the manuscript should be strengthened is in the framing about protected areas and their designation, which is established at the beginning of the Introduction. The authors should be clearer about the choices they make and definitions they employ in their paper and should set these in the context of other possibilities with regards to conservation decisions. Protected area networks may be established for a variety of reasons. One important reason is maintenance of biodiversity, which is the case this paper focuses on (L.55-57). But others are more species-specific or culturally driven (e.g., IUCN categories III - VI), which may lead to other criteria for effectiveness. It is fine for the paper to focus on biodiversity protection as a primary goal of protected area networks, but this should be explicitly stated and complemented with at least a sentence or two recognizing other purposes for protected areas. The paper focuses strongly on effectiveness. Statements such as that “A high degree of ecological representativeness is a necessary condition for an effective protected areas network” (L.69-70), that “These effects would increase the effectiveness of a protected areas network” (L.79-80), and that the generated set of protected areas “may be close to ideal from an environmental or conservation point of view” (L.345) assume a certain criterion for determining an effective protected areas network, ignoring other potential goals. Thus, biodiversity maintenance is presented as if it were the only possible definition for effectiveness, rather than one of multiple possible definitions. I recommend a more explicit statement early in the Introduction (e.g., ‘we define an “effective” regional protected areas network as one that can sustain the region’s biodiversity into some reasonably foreseeable future”) and discussion of other possible definitions. Some of this is touched on in the Discussion (e.g., L. 347-359) but it would be better to set this up in the Introduction as well.

OUR RESPONSE: We understand the interest in adding information to the manuscript in order to recognize the importance of other potential goals for the creation of protected areas. However, reporting all the goals for which we can protect land is beyond the scope of our manuscript. Regarding the importance of the biodiversity maintenance as one focus of our analysis, we do believe that this was already presented in the original manuscript, and still there in the revised version. The second sentence of the Introduction states: “Securing habitats by creating or expanding protected areas networks is part of the solution to the challenge of biodiversity loss [4].” We think this situates the study exactly as recommended by the reviewer. The third sentence of the introduction states: “A regional protected areas network could be considered ultimately effective insofar as it can sustain the region’s biodiversity into some reasonably foreseeable future.” reads to us effectively as a definition, as requested. 

MINOR COMMENTS:

COMMENT #53. L. 65-66. This assumes populations are stable or growing. The presence of species in ecological traps may artificially inflate representativeness, leading to assumption that “habitat requirements…will be satisfied within the protected areas network” when this is not actually the case. It might be nice to at least point out this possibility, even while acknowledging that this is nonetheless a typical assumption, as the paper does in L.68. In fact, this is a strange statement given the following paragraph, which details why it is not the assumption is not necessarily reasonable. It seems like it would make more sense for the paragraph on representativeness to make the case for why representativeness matters for conservation (e.g., see second paragraph of Dietz et al. 2015 Biological Conservation and references therein).

OUR RESPONSE: In the 1st paragraph of the Introduction we set up the thee aspects of protected areas network design that we address: representation, species-specific needs, climate change, which we then elaborate upon one at time. The second paragraph deals with the 1st of these, namely representation, includes the caveat “most species”. We have revised this paragraph to refer to make clear that by “habitat” we refer to or ecological types, citing Dietz et al. 2015, and changed “most” to “many”. The exceptions are discussed in the next paragraph, and we add another example, namely endangered species, as studied in Venter et al. 2014.

COMMENT #54. L.362. A citation is needed for this statement.

OUR RESPONSE: Rather than fully discuss and references these controversial points, we have simply dropped the two lines in question. The claims they make go beyond our results are not necessary to their interpretation. 

Low ecological representativeness may compromise the effectiveness or efficiency of protected areas networks. Ideally, managers would work towards achieving a better representation of the regional biodiversity when implementing new protected areas. However, to achieve biodiversity protection, protected areas networks should aim to represent the landscape in its pre-industrial form and not to protect habitat types resulting from human disturbances

COMMENT #55. L.366. Again, a citation is needed. In some cases, species thrive under habitat types resulting from human disturbances. It is a decision based on the sensitivity of focal species to human disturbance that should drive such decisions. 

OUR RESPONSE: Please see our response to Comment #54 above. 

COMMENT #56. L.376-377. This raises an important point about the potential to consider tradeoffs. This is not really done in the paper, however, as the focus is only on areas where both representativeness and functional connectivity are mutually high. One way this could be addressed is to make similar plots to Fig. 4 in the supplementary materials that are focused on just one of the indicators. Then the reader could better see where tradeoffs occur.

OUR RESPONSE: The focus not at all on [protected] areas where representativeness and functional connectivity are mutually high. It is on designing networks which, in aggregate, achieve both objectives. We discuss this in in the Introduction (lines 108-116) and in the section “Identifying priority conservation areas” (lines 311-324). The tradeoff curve could be shown explicitly by drawing a bounding curve over the clouds of points in Figures 3a and 3b; this could be done by quantile non-parametric additive models: but we feel this would obscure our main message which is that a better network designs are possible: better in both that representation and functional connectivity are improved relative to the Ministry’s proposal at that time.

GRAMMATICAL ISSUES TO ADDRESS:

L.81. “among the main current drivers” reads a bit awkwardly. Suggest replacing with “a major driver.” // OUR RESPONSE: Done.

L.90. Move comma within quotation marks. // OUR RESPONSE: Done.

For clarity and better readability, I recommend moving “Because of…existing networks” (L.89-91) to before “The effectiveness of…” in L.84. and deleting “One important…functional connectivity” (L.91-92). // OUR RESPONSE: Done.

L.111-121 are in present tense, L.122-134 in past tense. These should be consistent. // OUR RESPONSE: Done. We have put most such instances into the past tense.

L.152. “currently” is unnecessary. Delete. // OUR RESPONSE: Done.

L.212. Remove extra “s” in “areass." // OUR RESPONSE: Done.

L.252. Replace “or” with “of.” // OUR RESPONSE: Done.

L. 256. Remove “by simulating.” // OUR RESPONSE: Done.

L.269. Add an “s” to “subpopulation.” // OUR RESPONSE: Done.

L.292. Remove “the” between “protected areas” and “most frequently selected.” // OUR RESPONSE: Done.

L.374. Replace “ammeding” with “amending.” // OUR RESPONSE: Done.

L.418. Remove “to” after “benefit.” // OUR RESPONSE: Done.

L.427. Add “us” between “allows” and “to approximate.” // OUR RESPONSE: Done.

L.465. Add an “s” to “represent.” // OUR RESPONSE: Done.

Supporting Information

Last sentence of first paragraph. Replace “caution” with “cautious.” // OUR RESPONSE: Done.

Alpine tundra paragraph: 

o Consider replacing “for the horizon 2090” with “by 2090.” // OUR RESPONSE: Done.

o Add “using” between “…in Gaudry 2013)” and “interior buffering.” // OUR RESPONSE: Done.

o Consider adding “soils” between “serpentine” and “(Sirois and Grandtner 1992)” and changing “is less subject” to “are less subject.” // OUR RESPONSE: Done.

Mature fir stands and regenerating stands section / Impact of disturbances inside protected areas section:

o Consider replacing “turned” with “reclassified” in the second to last sentence. // OUR RESPONSE: Done.

Mature fir stands and regenerating stands section / Impact of disturbances outside protected areas:

o Remove “for” in “Windthrow is similarly managed for.” // OUR RESPONSE: Done.

o Replace “at” with “as” in “classed at mature.” // OUR RESPONSE: Done.

o Insert “were” between “Once the landscapes” and “built with the.” // OUR RESPONSE: Done.

o Consider rephrasing to something like “For the four climate change scenarios…the alpine tundra area was 100%, 59%, 39%, and 14%, respectively, compared to current conditions. This will indicate what the percentages indicate (a comparison with current). Also, “remained at” indicates stasis while the numbers reported reflect decreases in area. Similar phrasing should be used for changes in area of mature fir stands and regenerating stands. // OUR RESPONSE: Done.

o Consider deleting “fir stands area increased” as this seems to still be explaining why old stand area increased. // OUR RESPONSE: The first part of the sentence refers to OLD stands (not fir in particular) and the second part of the sentence refers to FIR stands (not particularly the old ones). We split the sentence in two for better clarity.

o Consider adding “regenerating” after “However,” as this sentence seems to be just about regenerating stands since old stands mostly increased. // OUR RESPONSE: The sentence referred to FIR stands, not regenerating stands. We coupled this part with the previous sentence that was split to show the two contrasting impacts on fir stands.

---

## [Decision Letter · Decision Letter 1]

9 Jul 2020

PONE-D-20-04608R1

Protected areas networks, functional connectivity and climate change: a caribou case-study

PLOS ONE

Dear Dr. St-Laurent,

Thank you for submitting your manuscript to PLOS ONE. After careful consideration, we feel that it has merit but does not fully meet PLOS ONE’s publication criteria as it currently stands. Therefore, we invite you to submit a revised version of the manuscript that addresses the points raised during the review process.

The reviewers are positive about revised paper, however, there is a need for some minor changes (methodology, figures, discussion).

We look forward to receiving your revised manuscript.

Kind regards,

Laurentiu Rozylowicz, Ph.D.

Academic Editor

PLOS ONE

Reviewers' comments:

Reviewer's Responses to Questions

**Comments to the Author**

1. If the authors have adequately addressed your comments raised in a previous round of review and you feel that this manuscript is now acceptable for publication, you may indicate that here to bypass the “Comments to the Author” section, enter your conflict of interest statement in the “Confidential to Editor” section, and submit your "Accept" recommendation.

Reviewer #2: (No Response)

Reviewer #3: (No Response)

2. Is the manuscript technically sound, and do the data support the conclusions?

Reviewer #2: Yes

Reviewer #3: Yes

3. Has the statistical analysis been performed appropriately and rigorously? 

Reviewer #2: Yes

Reviewer #3: Yes

4. Have the authors made all data underlying the findings in their manuscript fully available?

Reviewer #2: Yes

Reviewer #3: Yes

5. Is the manuscript presented in an intelligible fashion and written in standard English?

Reviewer #2: Yes

Reviewer #3: Yes

6. Review Comments to the Author

Reviewer #2: Thank you for your efforts to revise the manuscript in response to my previous review. The result is a clearer work that communicates the important approach you have taken. There remain, however, a few issues that should be addressed prior to acceptance of the manuscript. I hope that these will be easy to update and that the manuscript can then be accepted.

L.197-198. Both reviewers questioned giving an equal weight to each disturbance type, which suggests other readers will as well. Having read the authors’ responses to our comments, I now understand the authors’ reasoning behind this decision and suggest that a few sentences be added to clarify this for other readers. For example (adapting from the response to Comment 14): “…This gives equal weight to each disturbance type. While this is likely unrealistic for any given species, catchment intactness was intended to be relevant to biodiversity more broadly. Equal weighting reflects differing responses across species. For example, agricultural fields have a large impact on caribou but less impact on meso-carnivores (references). Even urban areas can be less avoided by some species (e.g., birds, insects) relative to either heavily forested areas or highways and their surroundings (references). Equal weighting thus represents a conservative assumption.”

I greatly appreciate the authors adding Fig 2 and making modifications to Fig 5 and the Methods description in response to my comments. This greatly helped in clarifying the approach used in their model. I still am a little confused, however, by what size threshold was used in creating candidate protected areas. In combination with Fig 2, L.209-213 indicate that starting with the seed catchments – which should be intact headwater catchments (presumably only on public land, see next point) – additional catchments along the stream network that met the intactness criteria of > median intactness were added sequentially until the target size of 85.5 sq km was met or slightly exceeded. At that point the set of combined catchments was considered a candidate protected area. The sentence from L.213-215 then seems redundant, simply restating what was just described so I suggest removing it. But then L.216-218 indicate that candidate protected areas with a total area above 31.2 sq km were selected. This is where I am confused. If the target size was 85.5 sq km and catchments were added until this was met or exceeded, what was the 31.2 sq km size used for? Should not all candidate protected areas have exceeded this size? I do not see this clearly explained. This needs to be clarified as the difference between these two target sizes affects the ability of readers to understand the criteria used and to replicate the study.

L.209. Fig 2 says intact headwater catchments on public land were used as seeds, but this line does not specify that the seeds had to be on public land. If this was a constraint it should be indicated here as well.

L.316,321. These references to Fig 2 seem a little strange to me as what is being described is not clearly pictured in Fig 2. There is some text that indicates seeking tradeoffs, but mostly I found myself looking for explanations of what was meant by “a subsample of 0.1% of the candidate networks that were highly ranked under all three criteria” and “using as the frequency threshold the inflection point in the proportional frequency curve,” neither of which seem to be indicated in Fig 2. I suggest removing these references to the figure as they really are not necessary here.

L.354. A reference is given to S3 Supporting Information to support the statement that “Most of these 501 designs did not exceed MELCCs functional connectivity scores, but some did.” However, MELCC is never mentioned in Supplementary Information S3, nor are specific values of the displayed protected areas or even a labelled color ramp of values given that would allow the reader to evaluate the claim made in the paper, so this reference seems inappropriate. Instead, S3 reports statistics based on a single optimality criterion only, which is not discussed anywhere in the main text. While I suggested making figures like this in my previous review, they need to be at least briefly introduced in the text and then given further description in the supplement, or else removed entirely.

L.396-399. I am unclear what is meant by “We attempted to address some of the potential shortcomings of such a coarse filter approach by amending a fine filter approach that accommodates very different criteria, which is builds in an almost orthogonal dimension to the conservation problem.” Which fine filter approach is meant? I first thought it was a reference to the caribou IBM, but considering that this is all under a heading of “Network ecological representativeness” and the caribou model fits under the next section, “Network functional connectivity,” I am left unclear by what the authors consider the fine filter approach? This is the only place in the manuscript where this term shows up other than the key words. What is meant by the fine filter and coarse filter approach need to be clearly described. Likewise, the potential shortcomings of the coarse filter approach should be described, at least in brief. Then it should be explained how the fine filter approach accommodates a different criteria and builds in an orthogonal dimension to the problem. As it stands, none of this is clear to me.

L.486. MDDELCC should be defined in the figure caption so that the figure can stand on its own. This is especially important because while the acronym MELCC is regularly used in the paper, MDDELCC does not show up anywhere in the main text or supplements, nor is there an MDDELCC 2014 reference in the References. The citation for this needs to be clarified.

L.505. Use of color in this figure improves clarity over the initial version. However, the greenscale color ramp used to show selection frequency of candidate protected areas in Figure 5 is subtle enough that it remains difficult to discern differences among many of the protected areas. It would be helpful to use a different color ramp with greater contrast. This also applies to the figures in Supplementary Information S3.

Finally, a number of typos or grammatical issues remain in the manuscript that should be addressed prior to publication. These include:

L.21. Change “disturbanes” to “disturbances”

L.133. Change “potental” to “potential”

L. 135. Change “and well as” to “as well as a”

L. 135. Remove extra space between “while” and “also”

L. 140. I suggest changing “has been” to “was”

L.167. Change “it” to “this caribou population” to clarify for those reading quickly that “it” does not refer to the park, as the subject of the previous sentence was Gaspésie National Park.

L.177. Change “completing” to “complementing”

L.271. Remove the comma from “model, process”

L.289. Remove the second “with” from “combined with with predicted”

L.438-439. Rephrase to “two genetically distinct sub-populations”

L.489. This should be changed to either “assembling a candidate protected areas network” or to “assembling candidate protected areas networks”. It was not clear to me which the authors intend.

L.491. Change “middel" to “middle”

L.491. Change “hydroligically” to “hydrologically”

L.496. Insert a comma after “connectivity”

Reviewer #3: ADDITIONAL COMMENTS IN THE EDITED PDF MANUSCRIPT

COMMENT #1: The research reported in this manuscript seems highly relevant to conservation planning for the Atlantic-Gaspésie caribou population. However, I think the new title of the manuscript should be changed a little bit. Hence, I suggest revising the title. A suggested title is “Integrating functional connectivity in designing networks of protected areas under climate change: a caribou case-study”.

COMMENT #2: L.348 I suggest you change the name “Identifying priority conservation areas” from the results to “Priority conservation areas” as it is the same as the title from the methods (L.310) and can be confusing.

COMMENT #3: L.504 “inflection” instead of “inflexion”

7. PLOS authors have the option to publish the peer review history of their article (what does this mean?). If published, this will include your full peer review and any attached files.

Reviewer #2: No

Reviewer #3: No

---

## [Author Response · Author response to Decision Letter 1]

21 Aug 2020

Integrating functional connectivity in designing networks of protected areas under climate change: a caribou case-study

(MS # PONE-D-20-04608.R2)

Responses to reviewers’ comments

Please find below our responses to the reviewer’s comments; the line numbers in our responses refer to the revised version (clean copy) of our manuscript.

Please note that we have also uploaded a version of our revised manuscript with all modifications highlighted in “Track Change”.

REVIEWER #2

COMMENT #1. Thank you for your efforts to revise the manuscript in response to my previous review. The result is a clearer work that communicates the important approach you have taken. There remain, however, a few issues that should be addressed prior to acceptance of the manuscript. I hope that these will be easy to update and that the manuscript can then be accepted.

OUR RESPONSE: Thanks for the time invested in reviewing our manuscript and helping us, your comments were greatly appreciated. 

COMMENT #2. L.197-198. Both reviewers questioned giving an equal weight to each disturbance type, which suggests other readers will as well. Having read the authors’ responses to our comments, I now understand the authors’ reasoning behind this decision and suggest that a few sentences be added to clarify this for other readers. For example (adapting from the response to Comment 14): “…This gives equal weight to each disturbance type. While this is likely unrealistic for any given species, catchment intactness was intended to be relevant to biodiversity more broadly. Equal weighting reflects differing responses across species. For example, agricultural fields have a large impact on caribou but less impact on meso-carnivores (Toews et al. 2018). Even urban areas can be less avoided by some species (e.g., birds, insects) relative to either heavily forested areas or highways and their surroundings. Equal weighting thus represents a conservative assumption.”

OUR RESPONSE: We have added some text to this section, adapted from your suggestions (lines 206-211). 

COMMENT #3. I greatly appreciate the authors adding Fig 2 and making modifications to Fig 5 and the Methods description in response to my comments. This greatly helped in clarifying the approach used in their model. I still am a little confused, however, by what size threshold was used in creating candidate protected areas. In combination with Fig 2, L.209-213 indicate that starting with the seed catchments – which should be intact headwater catchments (presumably only on public land, see next point) – additional catchments along the stream network that met the intactness criteria of > median intactness were added sequentially until the target size of 85.5 sq km was met or slightly exceeded. At that point the set of combined catchments was considered a candidate protected area. The sentence from L.213-215 then seems redundant, simply restating what was just described so I suggest removing it. But then L.216-218 indicate that candidate protected areas with a total area above 31.2 sq km were selected. This is where I am confused. If the target size was 85.5 sq km and catchments were added until this was met or exceeded, what was the 31.2 sq km size used for? Should not all candidate protected areas have exceeded this size? I do not see this clearly explained. This needs to be clarified as the difference between these two target sizes affects the ability of readers to understand the criteria used and to replicate the study.

OUR RESPONSE: We have revised the text to clarify and to motivate the distinction between the target and minimum sizes (Lines 218-227).

COMMENT #4. L.209. Fig 2 says intact headwater catchments on public land were used as seeds, but this line does not specify that the seeds had to be on public land. If this was a constraint it should be indicated here as well.

OUR RESPONSE: We specify in the preceding paragraph that “intact catchments” were defined so as to exclude catchments on private lands. We have modified Fig. 2 to remove the reference to this case-study-specific detail, so that the caption is now consistent with the text at the former L. 209. This avoids repeating the same detail in three places.

COMMENT #5. L.316,321. These references to Fig 2 seem a little strange to me as what is being described is not clearly pictured in Fig 2. There is some text that indicates seeking tradeoffs, but mostly I found myself looking for explanations of what was meant by “a subsample of 0.1% of the candidate networks that were highly ranked under all three criteria” and “using as the frequency threshold the inflection point in the proportional frequency curve,” neither of which seem to be indicated in Fig 2. I suggest removing these references to the figure as they really are not necessary here.

OUR RESPONSE: We agree and have removed the two references to Figure 2 (lines 325 and 327). This information actually refers more to Figure 3 but as Figure 3 presents results, we won’t refer to it in the Methods. The subsection in question (Lines 324-332) has been revised to improve clarity. 

COMMENT #6. L.354. A reference is given to S3 Supporting Information to support the statement that “Most of these 501 designs did not exceed MELCCs functional connectivity scores, but some did.” However, MELCC is never mentioned in Supplementary Information S3, nor are specific values of the displayed protected areas or even a labelled color ramp of values given that would allow the reader to evaluate the claim made in the paper, so this reference seems inappropriate. Instead, S3 reports statistics based on a single optimality criterion only, which is not discussed anywhere in the main text. While I suggested making figures like this in my previous review, they need to be at least briefly introduced in the text and then given further description in the supplement, or else removed entirely.

OUR RESPONSE: The material in S3 Supporting Information is indeed unrelated to the statement in question (lines 353-356 in the previous version). We now refer instead to Figure 3, which does support the statement (lines 368-370 of the revised version), and made also some revisions to the paragraph to improve notation and clarity (lines 365-384). With regards to S3, it is now motivated in the Methods (Lines 334-338), and references in the Results (Lines 379-384) and Discussion (Lines 470-472); the text of S3 itself has also been expanded, as requested

COMMENT #7. L.396-399. I am unclear what is meant by “We attempted to address some of the potential shortcomings of such a coarse filter approach by amending a fine filter approach that accommodates very different criteria, which is builds in an almost orthogonal dimension to the conservation problem.” Which fine filter approach is meant? I first thought it was a reference to the caribou IBM, but considering that this is all under a heading of “Network ecological representativeness” and the caribou model fits under the next section, “Network functional connectivity,” I am left unclear by what the authors consider the fine filter approach? This is the only place in the manuscript where this term shows up other than the key words. What is meant by the fine filter and coarse filter approach need to be clearly described. Likewise, the potential shortcomings of the coarse filter approach should be described, at least in brief. Then it should be explained how the fine filter approach accommodates a different criterion and builds in an orthogonal dimension to the problem. As it stands, none of this is clear to me.

OUR RESPONSE: The references to fine and course filter approaches were superfluous and have been deleted.

COMMENT #8. L.486. MDDELCC should be defined in the figure caption so that the figure can stand on its own. This is especially important because while the acronym MELCC is regularly used in the paper, MDDELCC does not show up anywhere in the main text or supplements, nor is there an MDDELCC 2014 reference in the References. The citation for this needs to be clarified.

OUR RESPONSE: Our use of “MDDELCC” as a citation key to Bouchard (2014) was incorrect. In the revised caption of Figure 1, we now use the journal’s numerical citation format. The acronym MDDELCC no longer appears in the caption or the Figure’s legend. We note that the responsible ministry has changed names several times since this project was initiated. We now refer throughout to “the Ministry”, as per the revised Introduction (Lines 161-162).

COMMENT #9. L.505. Use of color in this figure improves clarity over the initial version. However, the greenscale color ramp used to show selection frequency of candidate protected areas in Figure 5 is subtle enough that it remains difficult to discern differences among many of the protected areas. It would be helpful to use a different color ramp with greater contrast. This also applies to the figures in Supplementary Information S3.

OUR RESPONSE: We changed the color ramp for Figure 5 (Lines 529-532) and for the Figures in S3 Supporting Information.

COMMENT #10. Finally, a number of typos or grammatical issues remain in the manuscript that should be addressed prior to publication. These include:

• L.21. Change “disturbanes” to “disturbances”

OUR RESPONSE: Corrected.

• L.133. Change “potental” to “potential”

OUR RESPONSE: Corrected.

• L. 135. Change “and well as” to “as well as a”

OUR RESPONSE: Corrected.

• L. 135. Remove extra space between “while” and “also”

OUR RESPONSE: Corrected.

• L. 140. I suggest changing “has been” to “was”

OUR RESPONSE: Done.

• L.167. Change “it” to “this caribou population” to clarify for those reading quickly that “it” does not refer to the park, as the subject of the previous sentence was Gaspésie National Park.

OUR RESPONSE: Done.

• L.177. Change “completing” to “complementing”

OUR RESPONSE: Done.

• L.271. Remove the comma from “model, process”

OUR RESPONSE: Corrected.

• L.289. Remove the second “with” from “combined with with predicted”

OUR RESPONSE: Corrected.

• L.438-439. Rephrase to “two genetically distinct sub-populations”

OUR RESPONSE: Done.

• L.489. This should be changed to either “assembling a candidate protected areas network” or to “assembling candidate protected areas networks”. It was not clear to me which the authors intend.

OUR RESPONSE: This comment refers to the caption of Figure 2. Our intended meaning was the second you mentioned “assembling candidate protected areas networks”. The caption has been revised (Lines 512-515).

• L.491. Change “middel" to “middle”

OUR RESPONSE: Corrected.

• L.491. Change “hydroligically” to “hydrologically”

 OUR RESPONSE: Corrected.

• L.496. Insert a comma after “connectivity”

OUR RESPONSE: Corrected.

REVIEWER #3

COMMENT #11 – Title. The research reported in this manuscript seems highly relevant to conservation planning for the Atlantic-Gaspésie caribou population. However, I think the new title of the manuscript should be changed a little bit. Hence, I suggest revising the title. A suggested title is “Integrating functional connectivity in designing networks of protected areas under climate change: a caribou case-study”.

OUR RESPONSE: We have revised the title much as you suggested.

COMMENT #12. L.348 I suggest you change the name “Identifying priority conservation areas” from the results to “Priority conservation areas” as it is the same as the title from the methods (L.310) and can be confusing.

OUR RESPONSE: We have modified the section heading according to your suggestion (line 363).

COMMENT #13. L.504 “inflection” instead of “inflexion”.

OUR RESPONSE: Done (line 527)

---

## [Editor Report · Decision Letter 2]

25 Aug 2020

Integrating functional connectivity in designing networks of protected areas under climate change: a caribou case-study

PONE-D-20-04608R2

Dear Dr. St-Laurent,

We’re pleased to inform you that your manuscript has been judged scientifically suitable for publication and will be formally accepted for publication once it meets all outstanding technical requirements.

Kind regards,

Laurentiu Rozylowicz, Ph.D.

Academic Editor

PLOS ONE
---

## [Editor Report · Acceptance letter]

4 Sep 2020

PONE-D-20-04608R2 

Integrating functional connectivity in designing networks of protected areas under climate change: a caribou case-study 

Dear Dr. St-Laurent:

I'm pleased to inform you that your manuscript has been deemed suitable for publication in PLOS ONE. Congratulations! Your manuscript is now with our production department. 

Kind regards, 

on behalf of

Dr. Laurentiu Rozylowicz 

Academic Editor

PLOS ONE